# CORRELATED POLICY OPTIMIZATION IN MULTI-AGENT SUBTEAMS

**Dingyang Chen**[1]    **Jianing Ye**[2]    **Zhenyu Zhang**[1]    **Xiaolong Kuang**[1]

**Xinyang Shen**[1]    **Ozalp Ozer**[1]    **Chongjie Zhang**[2]    **Qi Zhang**[3]

[1]Amazon    [2]Washington University in St. Louis    [3]Worcester Polytechnic Institute
{dingerwa, zhenyuzh, xkkuang, xinyans, ozalpo}@amazon.com
{jianing.y, chongjie}@wustl.edu   qzhang9@wpi.edu

## ABSTRACT

In cooperative multi-agent reinforcement learning, agents often face scalability challenges due to the exponential growth of the joint action and observation spaces. Inspired by the structure of human teams, we explore subteam-based coordination, where agents are partitioned into fully correlated subgroups with limited inter-group interaction. We formalize this structure using Bayesian networks and propose a class of correlated joint policies induced by directed acyclic graphs . Theoretically, we prove that regularized policy gradient ascent converges to near-optimal policies under a decomposability condition of the environment. Empirically, we introduce a heuristic for dynamically constructing context-aware subteams with limited dependency budgets, and demonstrate that our method outperforms standard baselines across multiple benchmark environments.

## 1 INTRODUCTION

Cooperative multi-agent reinforcement learning (MARL) enables autonomous agents to jointly optimize a common objective and has been applied to domains such as traffic control (Chu et al., 2019), multi-robot coordination (Corke et al., 2005), and power grid management (Callaway & Hiskens, 2010). In many real-world scenarios, humans naturally organize into *subteams*, groups that exhibit tight internal coordination and limited external interaction, allowing for specialization and reduced communication complexity. Inspired by this, we explore how agents in cooperative MARL can also benefit from structured subteams that induce localized coordination and scalable learning.

Forming subteams reduces the effective dimensionality of the joint action and observation spaces within each group, alleviating the curse of dimensionality that plagues centralized coordination. Moreover, many tasks exhibit weak interdependencies across certain agent clusters, motivating the design of policies that encourage strong intra-group correlations while ignoring unnecessary global entanglements. For instance, in a distributed search-and-rescue mission, drones surveying separate regions need strong coordination within each team but limited communication across distant ones.

To model such structured correlations, we employ Bayesian networks (BNs), where a joint policy is factorized according to a directed acyclic graph (DAG) over agents. Agents within the same subteam are fully connected in the DAG, enabling expressive, correlated policies. Across subteams, no edges are introduced, effectively enforcing conditional independence. This structure allows us to capture meaningful dependencies without incurring the full complexity of unstructured joint policies. Our contributions are as follows:

- As a warm-up, we extend Chen & Zhang (2023) by establishing a convergence rate for tabular softmax BN policy gradient ascent under any fixed DAG, strengthening their asymptotic results.
- Our main theoretical results focus on a subclass of BNs where the agents can be partitioned into subteams, where agents select actions in a fully correlated manner within a subteam and independently in different subteams. Under a decomposability condition on the reward and transition

functions (subject to bounded errors), we prove that for such BNs regularized policy gradient ascent converges to a policy with bounded suboptimality. The bound hinges on the decomposition errors and the sizes of the subteams.

- Finally, we propose a heuristic to construct context-aware DAGs dynamically from local observations with a limit on the number of edges, relaxing the assumptions such as oracle value functions and global observability. We integrate this with deep multi-agent reinforcement learning algorithms and demonstrate that our method outperforms the state-of-the-art across several benchmark environments.

## 2 RELATED WORK

**Product policies in MARL.** MARL algorithms often adopt product policies, where the joint policy is represented as the product of agents' individual policies (Kuba et al., 2022; Yu et al., 2021; Lowe et al., 2017; Zhong et al., 2024; Foerster et al., 2018; Liu et al., 2024; Egorov & Shpilman, 2022; Li et al., 2024). This factorization is widespread in MARL due to its scalability and the ability to execute policies without communication at runtime. Despite their empirical success, policy gradient-based optimization methods for product policies are generally not guaranteed to converge to the global optimum (Ye et al., 2023). Most existing theoretical results focus on convergence to Nash equilibria, which is a weaker solution concept than global optimality (Leonardos et al., 2021; Chen et al., 2022; Ding et al., 2022; Fox et al., 2022; Sun et al., 2023).

**Correlated policies via Bayesian networks.** To address the suboptimality of product policies within the policy gradient framework, a number of works have proposed optimizing correlated joint policies, since a deeper correlation indicates a stronger expressiveness of the joint policy class, and thus often a better optimality guarantee. One popular approach for representing a correlated joint policy is to use Bayesian networks (BNs) (Heckerman, 2020). These methods (Ye et al., 2023; Chen & Zhang, 2023; Ruan et al., 2022; Christianos et al., 2023) represent the joint policy as a directed acyclic graph (DAG), allowing the joint policy to be factored into a product of several correlated conditional distributions. However, suboptimality persists whenever the BN is not fully connected.

**Value-based methods in Markov Team Problems.** There are also a number of works that have studied value-based methods in Markov Team Problems such as Littman (2001); Donmez et al. (2025); Sunehag et al. (2018); Rashid et al. (2018); Wang et al. (2020a). These studies typically treat the agent as a whole, tackling the scalability issue by the implicit decomposability of their value functions. In contrast, Phan et al. (2021a); Zang et al. (2023); Wang et al. (2020c;b); Kapoor et al. (2025) develop methods to explicitly group agents into subteams to learn a uncorrelated or factorized value function among subteams for better scalability. A more similar concept among value-based methods is the coordination graph (Guestrin et al., 2002; Böhmer et al., 2020; Li et al., 2020; Yang et al., 2022; Kang et al., 2022; Wang et al., 2022), which factorizes the joint value function according to a graph structure that encodes the coordination relationships among agents. Nonetheless, the optimality of these approaches is affected by the incompleteness of the function class and the imperfect approximation of the TD target under the imposed graph structure (Fioretto et al., 2016).

**Structural assumptions and sparse correlations.** For a policy with weak correlation (e.g., a sparse BN), the optimality of the algorithm may rely on certain assumptions about the environment. There is relatively limited theoretical work in this area. Some early research has demonstrated that if the transitions between agents exhibit some independence, certain algorithms (such as dynamic programming and independent learning) can achieve global optimality (Lauer & Riedmiller, 2004; Becker et al., 2004; Zhang & Lesser, 2011). Wang et al. (2021) and Dou et al. (2022) prove the convergence of the value-based algorithm VDN (Sunehag et al., 2018) under the assumption that the environment admits a decomposable structure. Building upon similar decomposability assumptions in Dou et al. (2022), our work extends the theoretical guarantees to the class of policy gradient based methods with BN represented correlated policies. To our best knowledge, this is the first work that establishes optimality guarantees for BN policies without requiring full independence among agents.

## 3 PRELIMINARIES

We consider a cooperative Markov game (MG) defined by tuple $\langle \mathcal{N}, \mathcal{S}, \mathcal{A}, P, r, \mu \rangle$, involving $N$ agents indexed by $i \in \mathcal{N} = \{1, \ldots, N\}$. The game consists of a state space $\mathcal{S}$, a joint action space $\mathcal{A} = \mathcal{A}^1 \times \cdots \times \mathcal{A}^N$ with $\mathcal{A}^i$ being the action space of agent $i$, a transition function $P : \mathcal{S} \times \mathcal{A} \to \Delta(\mathcal{S})$, a shared team reward function $r : \mathcal{S} \times \mathcal{A} \to \mathbb{R}$, and an initial state distribution $\mu \in \Delta(\mathcal{S})$. Here, $\Delta(\mathcal{X})$ denotes the set of probability distributions over $\mathcal{X}$. The game progresses in discrete time steps with next states and rewards generated from $P$ and $r$, respectively. The discounted cumulative reward from time step $t$ is denoted as $R_t := \sum_{l=0}^{\infty} \gamma^l r_{t+l}$ with $r_t := r(s_t, a_t)$. With full observability, meaning each agent observes the global state $s \in \mathcal{S}$, a general joint policy $\pi : \mathcal{S} \to \Delta(\mathcal{A})$ maps states to distributions over the joint action space, inducing its value function $V_\pi(s_t) := \mathbb{E}_{s_{t+1:\infty}, a_{t:\infty} \sim \pi}[R_t | s_t]$, the action-value function $Q_\pi(s_t, a_t) := \mathbb{E}_{s_{t+1:\infty}, a_{t+1:\infty} \sim \pi}[R_t | s_t, a_t]$, and its (unnormalized) discounted state visitation measure $d_\mu^\pi(s) := \mathbb{E}_{s_0 \sim \mu}[\sum_{t=0}^{\infty} \gamma^t \mathrm{Pr}^\pi(s_t = s | s_0)]$. The objective is to optimize the joint policy to maximize its value with respect to the initial state distribution, i.e., $\max_\pi V_\pi(\mu) := \mathbb{E}_{s_0 \sim \mu}[V_\pi(s_0)]$. Denote the optimal value as $V_*(\mu) := \max_\pi V_\pi(\mu)$. We say $\pi$ is $\epsilon$-*optimal* if its suboptimality

$$\mathrm{subopt}(\pi) := V_*(\mu) - V_\pi(\mu) \leq \epsilon.$$

Given the exponential growth of $\mathcal{A}$ with $N$, the commonly used joint policy subclass is the *product policy*, $\pi = (\pi^1, \ldots, \pi^N) : \mathcal{S} \to \times_{i \in \mathcal{N}} \Delta(\mathcal{A}^i)$, where joint policy $\pi$ is factored as a product of local policies $\pi^i : \mathcal{S} \to \Delta(\mathcal{A}^i)$, such that $\pi(a|s) = \prod_{i \in \mathcal{N}} \pi^i(a^i|s)$. It is well known that there exists a deterministic policy (and hence a product policy) with zero suboptimality.

Although the best product policy does not introduce suboptimality, the restriction of conditional independence among agents' actions restricts the expressiveness of the joint policy, which creates difficulties for optimizing the joint policy and often results in suboptimal behavior (Ye et al., 2023). Chen & Zhang (2023) extended beyond product policies by incorporating correlation in the local policies through a Bayesian network (BN) among the agents. A BN is represented by a directed acyclic graph (DAG) $G = (\mathcal{N}, \mathcal{E})$ with agents $\mathcal{N}$ being the vertices and $\mathcal{E} \subseteq \{(i,j) : i, j \in \mathcal{N}, i \neq j\}$ being the set of directed edges. The parents of an agent $i$ are denoted by $\mathcal{P}^i := \{j : (j,i) \in \mathcal{E}\}$ with their actions denoted as $a^{\mathcal{P}^i} \in \mathcal{A}^{\mathcal{P}^i} := \times_{j \in \mathcal{P}^i} \mathcal{A}^j$. DAG $G$ induces a joint policy $\pi_G = (\pi_G^1, \ldots, \pi_G^N) : \mathcal{S} \to \Delta(\mathcal{A})$, where each agent $i$'s local policy $\pi_G^i : \mathcal{S} \times \mathcal{A}^{\mathcal{P}^i} \to \Delta(\mathcal{A}^i)$ is conditioned on both the global state and the actions of its parents, and therefore joint action $a = (a^1, \ldots, a^N)$ is selected as $\pi_G(a|s) = \prod_{i \in \mathcal{N}} \pi_G^i(a^i|s, a^{\mathcal{P}^i})$. It is clear that BN policies define a continuum: $\pi_G$ reduces to a product policy when $G$ has no edges and is as expressive as general joint policies when $G$ is dense. We define the *equilibrium gap* of a BN policy as:

$$\mathrm{gap}^i(\pi_G) := \max_{\bar{\pi}_G^i} V_{\bar{\pi}_G^i, \pi_G^{-i}}(\mu) - V_{\pi_G}(\mu), \quad \mathrm{gap}(\pi_G) := \max_{i \in \mathcal{N}} \mathrm{gap}^i(\pi_G).$$

Here, the deviating BN policy $(\bar{\pi}_G^i, \pi_G^{-i})$ is consistent with $\pi_G$ in terms of the underlying $G$. We say BN policy $\pi_G$ is an $\epsilon$-approximate equilibrium if $\mathrm{gap}(\pi_G) \leq \epsilon$. Note this equilibrium notion resembles standard ones like Nash equilibrium (Nash, 1951) and coarse correlated equilibrium (Aumann, 1987), but they are different: Nash equilibrium only applies to product policies; the coarse correlated equilibrium applies to general policies but does not allow the deviating local policy to condition on any other agent's action.

**Shorthand notations.** When the underlying DAG is clear from the context, we will drop subscript $G$ and write a BN policy as $\pi$. For a subset $\mathcal{M} \subseteq \mathcal{N}$ of the agents and its complement $-\mathcal{M}$, a joint action is decomposed as $a = (a^{\mathcal{M}}, a^{-\mathcal{M}})$. The conditionals of policy $\pi$ given some $a^{-\mathcal{M}}$ is defined as $\pi(a^{\mathcal{M}}|s, a^{-\mathcal{M}}) := \pi(a^{\mathcal{M}}, a^{-\mathcal{M}}|s)/\sum_{\bar{a}^{\mathcal{M}}} \pi(\bar{a}^{\mathcal{M}}, a^{-\mathcal{M}}|s)$, with the corresponding action-value function $Q_\pi(s, a^{\mathcal{M}}) := \mathbb{E}_{a^{-\mathcal{M}} \sim \pi(\cdot|s, a^{\mathcal{M}})}[Q_\pi(s, a^{\mathcal{M}}, a^{-\mathcal{M}})]$. Let $\mathcal{P}_+^i := \mathcal{P}^i \cup \{i\}$ denote the set of agent $i$ and its parents.

## 4 WARM-UP: CONVERGENCE OF TABULAR BN POLICY GRADIENT ASCENT

Prior work (Chen & Zhang, 2023) established the asymptotic convergence of tabular softmax BN policy gradient ascent under any fixed DAG. To provide formality and as a warm-up, we here extend their result to get a finite-time convergence rate with the help of log barrier regularizer.

Fixing DAG $G$, we consider parameterizing local policies of a BN policy in the tabular softmax manner from the global state and parent actions as in Chen & Zhang (2023), i.e., for each agent $i$, we have its policy parameter and the induced softmax policy as

$$\theta^i = \left\{ \theta^i(s, a^{\mathcal{P}^i}, a^i) \in \mathbb{R} : s \in \mathcal{S}, a^{\mathcal{P}^i} \in \mathcal{A}^{\mathcal{P}^i}, a^i \in \mathcal{A}^i \right\}, \ \pi^i_{\theta^i}\left(a^i | s, a^{\mathcal{P}^i}\right) \propto \exp\left(\theta^i(s, a^{\mathcal{P}^i}, a^i)\right) \quad (1)$$

and the BN policy is therefore parameterized as $\pi_\theta = (\pi^1_{\theta^1}, \cdots, \pi^N_{\theta^N})$.

To provide a finite-time convergence guarantee, we optimize a log barrier regularized objective in a similar fashion to Agarwal et al. (2021) for the single-agent counterpart:

$$L_\lambda(\theta) := V_\theta(\mu) - \lambda \sum_{i \in \mathcal{N}} \mathbb{E}_{s, a^{\mathcal{P}^i} \sim \text{Unif}_{\mathcal{S} \times \mathcal{A}^{\mathcal{P}^i}}} \left[ \text{KL}\left( \text{Unif}_{\mathcal{A}^i}, \pi^i_{\theta^i}(\cdot | s, a^{\mathcal{P}^i}) \right) \right] \quad (2)$$

where $V_\theta$ and $Q_\theta$ are shorthands for $V_{\pi_\theta}$ and $Q_{\pi_\theta}$ in this paper; $\lambda > 0$ is the regularization parameter; $\text{Unif}_{\mathcal{X}}$ is the uniform distribution over $\mathcal{X}$; $\text{KL}(\cdot, \cdot)$ denotes the KL divergence. The log barrier regularization, i.e., the KL divergence with respect to the uniform action-selection distribution, is applied to each agent's policy independently. The standard gradient ascent for $L_\lambda(\theta)$ in (2) is

$$\theta^i_{t+1} = \theta^i_t + \eta \nabla_{\theta^i} L_\lambda(\theta_t) \qquad \forall i \in \mathcal{N} \quad (3)$$

where $\eta$ is a fixed stepsize. The explicit regularized policy gradient form is shown in Lemma 1.

**Lemma 1** (Proof in A.2). *For the BN policy parameterized as in Equation* (1)*, we have:*

$$\frac{\partial L_\lambda(\theta)}{\partial \theta^i(s, a^{\mathcal{P}^i}, a^i)} = \frac{1}{1-\gamma} d^{\pi_\theta}_\mu(s, a^{\mathcal{P}^i}) \pi^i_{\theta^i}(a^i | s, a^{\mathcal{P}^i}) A^i_\theta(s, a^{\mathcal{P}^i}, a^i) + \frac{\lambda}{|\mathcal{S}||\mathcal{A}^{\mathcal{P}^i}|} \left( \frac{1}{|\mathcal{A}^i|} - \pi^i_{\theta^i}(a^i | s, a^{\mathcal{P}^i}) \right)$$

*where* $d^{\pi_\theta}_\mu(s, a^{\mathcal{P}^i}) := d^{\pi_\theta}_\mu(s) \sum_{a^{-\mathcal{P}^i}} \pi_\theta(a^{-\mathcal{P}^i}, a^{\mathcal{P}^i} | s)$, $A^i_\theta(s, a^{\mathcal{P}^i}, a^i) := Q_\theta(s, a^{\mathcal{P}^i}_+) - Q_\theta(s, a^{\mathcal{P}^i})$.

The gradient form in Lemma 1 enables us to extend the single-agent finite-time convergence guarantees (Agarwal et al., 2021) to BN joint policies under the same assumptions used in the convergence results for product policies (Zhang et al., 2022; Chen et al., 2022), which we state below and are required in all of the theoretical results in this paper.

**Assumption 1.** *For any joint policy* $\pi$ *and any state* $s$ *of the Markov game,* $d^\pi_\mu(s) > 0$.

Assumption 1 is standard (e.g., Agarwal et al. (2021); Zhang et al. (2024; 2022); Chen & Zhang (2023)) and holds if the initial-state distribution satisfies $\mu(s) > 0$ for all $s \in \mathcal{S}$, ensuring every state is reachable with positive probability under any policy.

**Assumption 2.** *The reward function* $r$ *is bounded in the range* $[0, 1]$*, such that the value function is bounded as* $\forall s, \pi, 0 \leq V_\pi(s) \leq 1/(1-\gamma)$.

Let $M := \max_{\pi, \pi'} \left\| d^\pi_\mu / d^{\pi'}_\mu \right\|_\infty$ quantify the maximum pointwise ratio between state visitation measures induced by any two policies. By Assumption 1, $M$ is well-defined and finite. This constant appears in Lemma 2 that extends the results in the single-agent setting (Agarwal et al., 2021) and the multi-agent setting with product policies (Zhang et al., 2022; Chen et al., 2022), stating that, with the log barrier, approximate first-order stationary points are approximate equilibria.

**Lemma 2** (Proof in A.3). *If* $\theta$ *is such that* $\|\nabla_\theta L_\lambda(\theta)\|_2 \leq \lambda/(2|\mathcal{S}||\mathcal{A}| \max_i |\mathcal{A}^i|)$*, BN policy* $\pi_\theta$ *is a* $2\lambda M$*-approximate equilibrium.*

With Lemma 2, we establish the convergence rate as stated in Theorem 1.

**Theorem 1** (Proof in A.4). *For any* $\epsilon > 0$*, under updates* (3) *beginning with* $\theta_0 = 0$ *and using* $\lambda = \frac{\epsilon}{2M}$ *and stepsize* $\eta \leq \frac{1}{\beta_\lambda}$ *with* $\beta_\lambda = \frac{8N}{(1-\gamma)^3} + \frac{2\lambda N}{|\mathcal{S}|}$ *being an upper bound on the smoothness of* $L_\lambda(\theta)$*, we have* $\min_{t \leq T} \texttt{gap}(\pi_{\theta_t}) \leq \epsilon$ *whenever*

$$T \geq \frac{256 N M^2 |\mathcal{S}|^2 \max_i |\mathcal{A}^i|^2}{(1-\gamma)^4 \epsilon^2} + \frac{32 N M |\mathcal{S}| \max_i |\mathcal{A}^i|^2}{(1-\gamma)\epsilon}. \quad (4)$$

The key idea in our proof is to reinterpret parent actions $a^{\mathcal{P}^i}$ as part of the state for agent $i$, treating the tuple $(s, a^{\mathcal{P}^i})$ as an augmented state. Under this formulation, the joint distribution $d^{\pi_\theta}_\mu(s, a^{\mathcal{P}^i})$ becomes the state visitation measure over the augmented state space. This transformation brings

the gradient ascent updates (3) into close alignment with those for the product policy (Zhang et al., 2022; Chen et al., 2022), enabling a natural generalization of the analysis to the BN policy setting.

Although Theorem 1 establishes a convergence rate to an approximate equilibrium, which is the strongest type of result one can expect for general cooperative MGs, it may still yield arbitrarily suboptimal policies. To address this, we next characterize a subclass of MGs where optimality can be guaranteed via regularized policy gradient ascent, along with an approximation where the suboptimality is explicitly quantified by deviations from the defining conditions for this subclass.

## 5 CONVERGENCE TO NEAR-OPTIMALITY VIA SUBTEAMS DECOMPOSITION

If the transition and reward functions of a cooperative MG can be decomposed into components associated with disjoint subsets of agents, then strong dependencies exist among the agents within each subset while agents in different subsets exhibit limited dependencies. In such a case, if the BN policy only preserves full correlation in the local policies within each subset (but not between the subsets), it turns out that the regularized policy gradient ascent can achieve near-optimal coordination, as we will establish in this section. As a first step, we define our notion of a *subteam*:

**Definition 1** (Subteam). *Given DAG $G = (\mathcal{N}, \mathcal{E})$ and a subset of its vertices (i.e., agents) $\mathcal{C} \subseteq \mathcal{N}$. The subgraph of $G$ induced by $\mathcal{C}$ is denoted as $G_{\mathcal{C}} = (\mathcal{C}, \mathcal{E}_{\mathcal{C}})$ with $\mathcal{E}_{\mathcal{C}} := \{(i, j) : i, j \in \mathcal{C}, (i, j) \in \mathcal{E}\}$. Subset $\mathcal{C}$ is a* subteam *in $G$ if, for every pair of distinct $i, j \in \mathcal{C}$, either $(i, j)$ or $(j, i)$ is in $\mathcal{E}_{\mathcal{C}}$.*

By Definition 1, agents in a subteam are fully connected by directed edges, subject to the the acyclicity constraint. For example, any single agent is a subteam; any pair of two connected agents is a subteam. Intuitively, for a BN policy with $G$ being the underlying DAG, the local policies in a subteam of $G$ are fully correlated in the sense that the BN policy is expressive enough to represent any joint action distribution of the agents in the subteam. We will partition all agents into subteams for a BN policy, which is reasonable when the cooperative MG of interest can be well decomposed by this partition. We define a cooperative MG's *decomposability by a partition of its agents* as follows:

**Definition 2** (Decomposition of a cooperative MG by a partition of agents). *Consider a cooperative MG $\langle \mathcal{N}, \mathcal{S}, \mathcal{A}, P, r, \mu \rangle$ and a collection of $K$ subsets of agents $\mathcal{N}$, $\{\mathcal{C}_k\}_{k=1}^K$, being a partition of $\mathcal{N}$, i.e., $\bigcup_{k=1}^K \mathcal{C}_k = \mathcal{N}$ and $\mathcal{C}_k \cap \mathcal{C}_{k'} = \emptyset \; \forall 1 \le k \ne k' \le K$. The MG is* decomposed by the partition *with errors $(\epsilon_P, \epsilon_r)$ if its transition function $P$ and reward function $r$ can be decomposed as*

$$P(s'|s, a) = \sum_{k=1}^K P^k(s'|s, a^{\mathcal{C}_k}) + \epsilon_P(s'|s, a), \quad r(s, a) = \sum_{k=1}^K r^k(s, a^{\mathcal{C}_k}) + \epsilon_r(s, a) \quad (5)$$

*for any $s, s' \in \mathcal{S}, a \in \mathcal{A}$ and some real-valued functions $\epsilon_P$, $\epsilon_r$, and $\{P^k, r^k\}_{1 \le k \le K}$.*

In words, the transition/reward function is decomposed into components, one per subset of the partition, where each component depends on actions taken by only the agents in the corresponding subset. We here make a few remarks on Definition 2: 1) We do not impose any regularity assumptions on $P^k$ and $r^k$; especially, $P^k$ needs not be a probability measure. 2) Because the errors $(\epsilon_P, \epsilon_r)$ can be arbitrarily chosen, the decomposition of $P$ and $R$ given in Equation (5) is always feasible for any partition $\{\mathcal{C}_k\}_{k=1}^K$ of $\mathcal{N}$, as one can simply accommodate the decomposition errors into $(\epsilon_P, \epsilon_r)$. As one might expect, our suboptimality guarantee will degrade as the errors increase. Letting $|\epsilon_P| := \max_{s,a,s'} |\epsilon_P(s'|s, a)|$ and $|\epsilon_r| := \max_{s,a} |\epsilon_r(s, a)|$, we have the following proposition confirming the intuition that finer partitions only introduce larger decomposition errors.

**Proposition 1** (Proof in A.7). *Suppose $\{\mathcal{C}_k\}_{k=1}^K$ and $\{\mathcal{C}'_{k'}\}_{k'=1}^{K'}$ are two partitions of $\mathcal{N}$, and the latter is finer than the former in the sense that, for all $1 \le k' \le K'$, $\mathcal{C}'_{k'} \subseteq \mathcal{C}_k$ for some $1 \le k \le K$. If the MG is decomposed by $\{\mathcal{C}'_{k'}\}_{k'=1}^{K'}$ with errors $(\epsilon'_P, \epsilon'_r)$, then the MG can be decomposed by $\{\mathcal{C}_k\}_{k=1}^K$ with errors $(\epsilon_P, \epsilon_r)$ such that $|\epsilon_P| \le |\epsilon'_P|$ and $|\epsilon_r| \le |\epsilon'_r|$.*

We are ready to state the conditions that a DAG needs to satisfy for our main theoretical result:

**Assumption 3.** *For the cooperative MG of interest equipped with DAG $G = (\mathcal{N}, \mathcal{E})$, there is a collection of subsets $\{\mathcal{C}_k\}_{k=1}^K$ satisfy the following conditions:*

(i) *$\{\mathcal{C}_k\}_{k=1}^K$ is a partition of $\mathcal{N}$; each $\mathcal{C}_k$ is a subteam in $G$ for all $1 \le k \le K$;*

(ii) *The MG is decomposed by $\{\mathcal{C}_k\}_{k=1}^K$ with errors $(\epsilon_P, \epsilon_r)$;*

*(iii) For any $1 \leq k \neq k' \leq K$, $\mathcal{E}$ does not have any edge $(i, j)$ for $i \in \mathcal{C}_k$ and $j \in \mathcal{C}_{k'}$.*

Conditions (i) and (ii) are directly taken from Definitions 1 and 2. Condition (iii) excludes any edge between any two subsets. While any additional edges increase the expressiveness of the induced BN policy and therefore should ease the policy optimization, (iii) is technically required in our proof. Specifically, a key step in Lemma 3's proof is to upper bound the gain of $a_k^c$ over the subteam baseline $V_\theta^k(s)$, which derived from a telescoping sum that would fail if subteams are not independent, as shown in the proof in Appendix A.5.

In the remainder of this section, we consider the regularized policy gradient ascent (3) to optimize the tabular softmax BN policy induced by a DAG that satisfies the conditions in Assumption 3 for a given cooperative MG. The following lemma states that, in this case, the approxmiate equilibrium guarantee in Lemma 2 can be strengthened into a near-optimality one. For ease of exposition, define $g(\{\mathcal{C}_k\}_{k=1}^K) := \sum_{k=1}^K 2^{|\mathcal{C}_k|} - K$, where $g(\cdot)$ is a real-valued function of an arbitrary collection of sets $\{\mathcal{C}_k\}_{k=1}^K$; its output value depends on the number and the sizes of the sets.

**Lemma 3** (Proof in A.5). *Suppose BN policy $\pi_\theta$ is parameterized in the tabular softmax manner as in Equation (1) with the underlying DAG satisfying all conditions in Assumption 3 with partition $\{\mathcal{C}_k\}_{k=1}^K$. If $\theta$ is such that $\|\nabla_\theta L_\lambda(\theta)\|_2 \leq \lambda/(2|\mathcal{S}||\mathcal{A}| \max_i |\mathcal{A}^i|)$, $\pi_\theta$ is*

$$\left( 2\lambda M g(\{\mathcal{C}_k\}_{k=1}^K) + 2K\left( \frac{|\epsilon_r|}{(1-\gamma)} + \frac{\gamma|S||\epsilon_P|}{(1-\gamma)^2} \right) \right) \text{- optimal.} \tag{6}$$

Compared with Lemma 2, the requirement on $\theta$ being an approximate stationary point remains the same, yet the bound of the equilibrium gap is strengthen to a suboptimality bound in (6) consisting of two terms. The first term can be made arbitrarily small by choosing a sufficiently small regularization parameter $\lambda$ like in Lemma 2 but also quantifies the impact of the partition with function $g(\cdot)$, the value of which becomes smaller as the partition gets finer as stated in the following proposition:

**Proposition 2** (Proof in A.8). *Suppose $\{\mathcal{C}_k\}_{k=1}^K$ and $\{\mathcal{C}'_{k'}\}_{k'=1}^{K'}$ satisfy the same conditions in Proposition 1. We have $g(\{\mathcal{C}'_{k'}\}_{k'=1}^{K'}) \leq g(\{\mathcal{C}_k\}_{k=1}^K)$.*

The second term captures the impact of the decomposition errors, which increases as the decomposition errors become larger as stated in Proposition 1. Therefore, the suboptimality bound in (6) reveals a tradeoff when choosing the fineness/coarseness of the decomposition.

In a similar manner, the guarantee of finite-time convergence to an approximate equilibrium in Theorem 1 can be strengthened into a near-optimality one as stated below.

**Theorem 2** (Proof in A.6). *Suppose the underlying DAG $G$ of BN policy $\pi_\theta$ satisfies the same conditions as in Lemma 3. For any $\epsilon > 0$, under updates (3) beginning with $\theta_0 = 0$ and using $\lambda = \frac{\epsilon}{2} M^{-1} g(\{\mathcal{C}_k\}_{k=1}^K)^{-1}$ and $\eta \leq \frac{1}{\beta_\lambda}$ ($\beta_\lambda$ as in Theorem 1), we have*

$$\min_{t \leq T} \text{subopt}(\pi_{\theta_t}) \leq \epsilon + 2K\left( \frac{|\epsilon_r|}{(1-\gamma)} + \frac{\gamma|S||\epsilon_P|}{(1-\gamma)^2} \right) \tag{7}$$

*whenever*

$$T \geq \frac{256NM^2|\mathcal{S}|^2|\mathcal{A}|^2 \max_i |\mathcal{A}^i|^2}{\epsilon^2(1-\gamma)^4} g(\{\mathcal{C}_k\}_{k=1}^K)^2 + \frac{32NM|\mathcal{S}||\mathcal{A}|^2 \max_i |\mathcal{A}^i|^2}{\epsilon(1-\gamma)} g(\{\mathcal{C}_k\}_{k=1}^K). \tag{8}$$

The second term in (7) matches the second term of (6) from Lemma 3, which quantifies the asymptotic suboptimal bias caused by the decomposition errors. In the extreme case of $K = 1$, all agents form a single subteam, making the decomposition error-free ($|\epsilon_P| = |\epsilon_r| = 0$) and ensuring $\epsilon$-optimality. A larger $K$ tries to impose stronger independence assumptions across the subteams, often resulting in larger $|\epsilon_P|$ and $|\epsilon_r|$ and therefore a larger asymptotic suboptimal bias. Regarding the convergence rate of (8), the dominating term is the first one that scales with $1/\epsilon^2$, which matches the first term of (4) from Theorem 1 up to the factor of $g(\{\mathcal{C}_k\}_{k=1}^K)^2$ that favors finer partitions into subteams according to Proposition 2. Increasing $K$ can create finer partitions and speed up the convergence. However, this speedup comes at the cost of potentially larger decomposition errors and thus greater suboptimality. This reveals a fundamental trade-off in subteams design.

**Proof sketch.** We here provide the key steps in our proof of Theorem 2. *1) Value function decomposition:* By Definition 2, the transition function and reward function decompose by the subteams $\{\mathcal{C}_k\}_{k=1}^K$ with additive errors $\epsilon_P$ and $\epsilon_r$. Consequently, the global value functions admit additive decompositions: $Q_\theta(s, a) = \sum_{k=1}^K Q_\theta^k(s, a^{\mathcal{C}_k}) + \epsilon_{Q_\theta}(s, a), V_\theta(s) = \sum_{k=1}^K V_\theta^k(s) + \epsilon_{V_\theta}(s),$

where error terms $\epsilon_{Q_\pi}$ and $\epsilon_{V_\pi}$ accumulate from errors $(\epsilon_P, \epsilon_r)$. *2) Bounding subteam advantage:* From Lemma 1, the regularized policy gradient provides a bound on local advantage $A_\theta^i(s, a^{\mathcal{P}^i}, a^i)$. Consider a topological ordering over the agents in a subteam $\mathcal{C}_k$. For agents $i$ and $j$ such that $j$ directly precedes $i$ in the ordering, we have $a^{\mathcal{P}^i} = a^{\mathcal{P}^j_+}$ due to the full connectivity in $\mathcal{C}_k$ and no cross-subteam connectivity, implying $Q_\theta(s, a^{\mathcal{P}^i}) = \mathbb{E}_{\bar{a}^{-\mathcal{P}^j_+} \sim \pi_\theta(\cdot|s, a^{\mathcal{P}^j_+})} \left[ Q_\theta(s, a^{\mathcal{P}^j_+}, \bar{a}^{-\mathcal{P}^j_+}) \right] = Q_\theta(s, a^{\mathcal{P}^j_+})$. Applying this in the reverse topological order and combining with the local advantage bound yields a bound on $Q_\theta(s, a^{\mathcal{C}_k}) - V_\theta(s)$. *3) Bounding $Q_\pi^k(s, a^{\mathcal{C}_k}) - V_\pi^k(s)$:* Apply the value function decomposition from step 1 to separate $Q_\theta(s, a^{\mathcal{C}_k})$ and $V_\theta(s)$ into components dependent on $a^{\mathcal{C}_k}$ and $a^{-\mathcal{C}_k}$. Their difference cancels out unrelated components, producing a bound on $Q_\pi^k(s, a^{\mathcal{C}_k}) - V_\pi^k(s)$ up to approximation errors that can be bounded using errors $(\epsilon_P, \epsilon_r)$. *4) Lemma 3 and the finite-time convergence:* Summing the bounds from the previous step across all subteams gives a bound on $Q_\theta(s, a) - V_\theta(s)$, which leads to the suboptimality guarantee as stated in Lemma 3. Using the same convergence argument as in Theorem 1, we choose a sufficiently small stepsize $\eta$ for gradient ascent. Since $L_\lambda(\theta)$ is smooth, we can guarantee that after enough iterations, the gradient norm becomes small enough to invoke Lemma 3. With a proper choice of $\lambda$, this leads to the statement in Theorem 2.

# 6 EMPIRICAL RESULTS

Our empirical study progresses in two parts. In Section 6.1, we begin with experiments that exactly adhere to the setting in Section 5, providing an empirical analysis of our theoretical results. Next, informed by the theoretical insights, in Section 6.2 we propose a practical heuristic that constructs subteams that potentially induce low decomposition errors given an edge budget, which is integrated into and improves state-of-the-art deep MARL algorithms.

## 6.1 TABULAR EXACT GRADIENT ASCENT WITH FIXED DAG IN THE COORDINATION GAME

We consider an $N$-agent extension of the two-player Coordination Game in Zhang et al. (2024) with $N \in \{2, 3, 5\}$. Each agent has a binary local state and action space, $\mathcal{S}^i = \mathcal{A}^i = \{0, 1\}$. The reward function encourages agents to align their local states, with a preference for global configurations containing more agents in state 0 when majority counts are tied. The local state transition of agent $i$ depends only on its own action: $P(s^i = 0|a^i = 0) = 1 - \epsilon$, $P(s^i = 0|a^i = 1) = \epsilon$, where $\epsilon = 0.05$.

We intentionally choose this minimal and didactic domain, so that we are able to afford the requirements of the theoretical results: exact gradient ascent for the tabular softmax BN policy parameterization with a fixed DAG. We compare the following DAG topologies: 1) the `product` DAG with no edges; 2) the `full` DAG where every pair of agents is connected, so it is $K = 1$ subteam including all agents; 3) DAGs with $K$ subteams and have no edges between any two subteams are labeled with the subteam sizes, e.g., `2+3` for $N = 5$ agents; 4) the `line` DAG where each agent $i < N$ is connected to agent $i + 1$, as considered in prior work (Böhmer et al., 2020; Chen & Zhang, 2023). Note that all DAGs satisfy Assumption 3, except for the `line` DAG.

Figure 4 in the appendix confirms that all DAGs converge to equilibria, in agreement with Theorem 1. We assess the optimality gap across subteam partitions in Figure 1: For $N = 2, 5$, policies with coarser partitions (i.e., fewer, larger subteams) consistently achieve higher final performance. For example, with $N = 5$, `full` performs similarly to `1+4` and are the best, followed by `2+3`, and finally the `product`. For $N = 3$, `full` still performs the best, while `line` and `product` perform similarly, with `1+2` slightly worse.

To explain this ordering, we fit $\{P^k, r^k\}_{k=1}^K$ with three-layer multilayer perceptrons by minimizing the decomposition errors when regressed to the transition and reward functions (cf. Definition 2). Table 1 presents the fitted errors across subteam partitions. Notably, the partitions that yield smaller decomposition errors consistently cor-

Table 1: Fitted decomposition errors.

| $N$ | DAG | $|\epsilon_P|$ | $|\epsilon_r|$ |
|---|---|---|---|
| 2 | `product` | 2.04e-01 | 1.26e+00 |
| | `full` | 3.57e-03 | 2.38e-07 |
| 3 | `1+2` | 2.94e-01 | 1.50e+00 |
| | `product` | 3.96e-01 | 2.00e+00 |
| | `full` | 2.21e-05 | 2.79e-09 |
| 5 | `1+4` | 3.38e-01 | 1.44e+00 |
| | `2+3` | 4.80e-01 | 2.38e+00 |
| | `product` | 6.03e-01 | 1.56e+00 |
| | `full` | 7.13e-08 | 3.73e-08 |

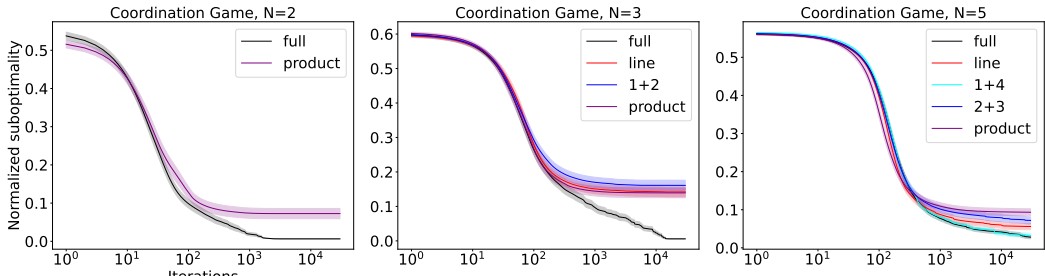

Figure 1: Results of tabular softmax BN policy gradient ascent under various DAG topologies. Averaged over 50 seeds, with shaded areas showing standard error; normalized suboptimality at iteration $t$ is defined as $1 - V_{\theta_t}(\mu)/V_*(\mu)$ and initial policy $\theta_0 \sim \mathcal{N}(0, 1)$.

respond to better-performing policies for most cases, which aligns with Theorem 2. The only exception is $N = 3$, where `1+2` has smaller errors `product`, but the product performs slightly better. A possible reason is that the additional correlation introduced by `1+2` may not yet be significant enough to yield a performance gain over `product`. Meanwhile, `product`, with fewer parameters, may be easier to optimize and thus achieves better performance in this particular case.

## 6.2 PRACTICAL METHODS

We have been focusing on exact gradient updates of (3) to optimize BN policies with fixed DAGs. Further, while our results so far justify partitions that induce low decomposition errors, finding such partitions is non-trivial. These issues motivate us to introduce below a heuristic approach that dynamically constructs subteams in a way that potentially induces low decomposition errors given an edge budget. This method can be easily integrated into deep MARL algorithms for practical use.

**Dependency-based subteams construction.** Given a limit of at most $B$ edges, our heuristic method constructs a DAG, along with a collection of subteams in the DAG that satisfies the conditions in Assumption 3. The construction is guided by dependency scores, $\{d_{ij}\}_{i,j \in \mathcal{N}}$, which are given a priori based on domain knowledge that roughly quantify the dependency between any pair of agents. Initialized with singleton subteams with no edges, the core idea is to iteratively merge two subteams $\mathcal{C}, \mathcal{C}'$ that maximize the average pair-wise dependency score between the agents in the two subteams: $d(\mathcal{C}, \mathcal{C}') := \frac{1}{|\mathcal{C}||\mathcal{C}'|} \sum_{i \in \mathcal{C}, j \in \mathcal{C}'} d_{ij}$. The chosen two subteams are merged by adding edges between them, and the merging will repeat until reaching edge limit $B$, as outlined in Algorithm 1 in the appendix. Because merging larger subteams needs more edges, the averaging encourages efficiently use of the edge budget. The dependency scores can change dynamically, e.g., based on the state/episode information, to minimize decomposition errors in a context-aware manner.

### 6.2.1 BN POLICY WITH DYNAMIC DAG IN DEEP MULTI-AGENT ACTOR-CRITIC

For practical usage, we integrate the BN policy as the actor into deep multi-agent actor-critic algorithms such as MAPPO (Yu et al., 2022) and MADDPG (Lowe et al., 2017). During training, parent actions are detached from the computation graph to prevent backpropagation, which we find ensures proper credit assignment and stabilizes training. To handle the variable number of parent actions induced by the dynamic DAG, we construct a fixed-length input vector of size $N \cdot \sum_{i \in \mathcal{N}} |\mathcal{A}^i|$, where the actions of non-parent agents are zero-padded. This design enables consistent input formatting across different DAG topologies and supports efficient batch processing. The implementation details are provided in Appendix B.

**Environments and their dependency score.** For the *Coordination Game*, we treat each agent's local binary state as its 1D position, enabling a natural way to compute pairwise dependency scores based on positional proximity. We consider two more environments. *Aloha* from Wang et al. (2022) involves 10 agents arranged in a $2 \times 5$ grid, each maintaining a message queue and chooses whether

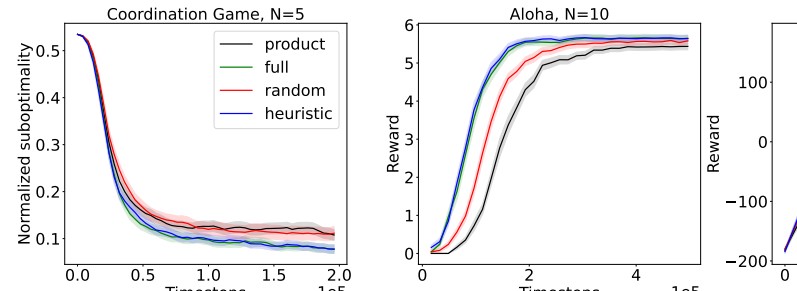

Figure 2: Results of integrating dynamic decomposition with MAPPO/MADDPG. Averaged over 60, 60, and 10 seeds for Coordination Game, Aloha, and Predator-Prey, respectively, with shaded areas showing standard error.

to transmit at each timestep. With probability 0.6, a new message is added to each queue at every step. A successful, collision-free transmission yields a global reward of 0.1, while a collision incurs a shared penalty of $-10$. We use Manhattan distances between the agents to define their dependency scores. *Predator-Prey* from Li et al. (2020) has $N = 15$ controllable predators and multiple uncontrollable preys moving in a 2D space. The environment introduces additional challenges compared to the previous ones, including stochastic initial positions and higher coordination complexity due to continuous movement. The dependency scores we define again rely on predators' spatial locations.

**Base algorithms, DAGs.** We select MAPPO as the base algorithm for Coordination Game and Aloha as they involve discrete action spaces. Predator-Prey involves continuous action spaces, so we adopt MADDPG. We compare four types of DAG topologies: `full` and `product` as described previously, the dynamic DAG constructed via our `heuristic` approach, and the `random` DAG. For fair comparison, `heuristic` and `random` are constrained under the same edge budget $B$ with $B = 4, 10, 50$ for Coordination Game, Aloha, and Predator-Prey, respectively.

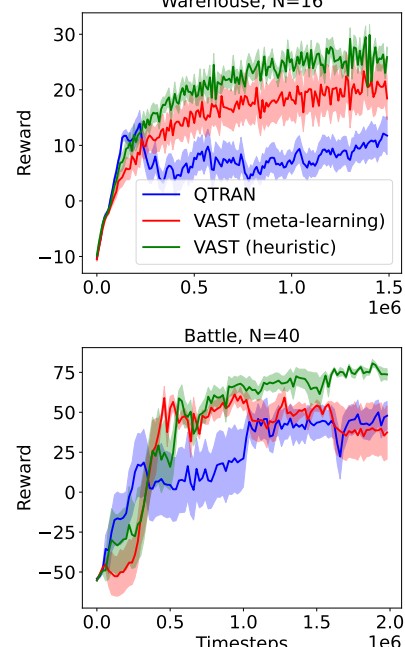

Figure 3: Our heuristic for VAST. Averaged over 10 and 5 seeds for Warehouse and Battle, respectively, with shaded areas showing standard error.

**Results.** As shown in Figure 2, our `heuristic` achieves the highest performance in all three environments. In Coordination Game, the performance is ordered as `heuristic` $\approx$ `full` > `random` $\approx$ `product`. In Aloha, while the four methods are comparable by end of training, `heuristic` and `full` clearly learns fastest. In Prey, `heuristic` is still best while `full` becomes the worst, with `random` comparable to `product`.

### 6.2.2 VALUE FACTORIZATION
PER AGENT SUBGROUPS IN CENTRALIZED TRAINING

We here repurpose our heuristic in Section 6.2 for centralized training methods that involves value factorization per agent subgroups. Specifically, we consider VAST (Phan et al., 2021b): given partition $\{\mathcal{C}_k\}_{k=1}^{K}$, VAST replaces agent-wise values as in traditional works like QTRAN (Son et al., 2019) with subgroup-wise ones $\{Q_\pi^k(s, a^{\mathcal{C}_k})\}_{k=1}^{K}$ and these values are fed into the mixer to estimate joint value $Q_\pi(s, a)$ as $Q_{\text{VAST}}(s, a) = \Phi_\psi\left(Q^1(s, a^{\mathcal{C}_1}), \ldots, Q^K(s, a^{\mathcal{C}_K})\right)$, where $\Phi_\psi$ is the mixer parameterized by $\psi$. Phan et al. (2021b)

consider various methods to determine subgroups $\{\mathcal{C}_k\}_{k=1}^K$ and their meta-learned approach performs the best. Our VAST variant instead determines $\{\mathcal{C}_k\}$ by our heuristic, with the mixer and learning losses follow the original VAST algorithm. We use the same edge budget in Phan et al. (2021b), which sets $K = \lceil \eta N \rceil$ with $\eta = 1/4$. Notably, like QTRAN, VAST falls into the centralized training and decentralized execution (CTDE) paradigm, which equivalently employs product policies with no inter-agent correlation. This is a fundamental difference from the previous sections of this paper.

**Environments and their dependency score.** *Warehouse:* $N$=16 robots move on a grid with shelves and stations. The objective is to pick items and deliver them efficiently. Rewards are positive for successful deliveries and include small penalties for wasted moves or blocking. *Battle:* $N$=40 units move and attack on a grid against forty opponents. The objective is to win local fights and advance. Rewards are positive for damaging or defeating enemies and negative for losses or ineffective actions. In both tasks dependency scores are computed from 2D positions as in Predator-Prey.

**Results.** Figure 3 shows that, in both Warehouse and Battle, VAST with our heuristic outperforms original VAST with their meta learning approach to determine the subgroups, which is the best variant reported in Phan et al. (2021b), and both VAST variants surpass the ungrouped QTRAN.

## 7 CONCLUSION

Our theoretical results establish finite-time convergence and suboptimality guarantees for BN policy gradient methods under decomposability assumptions on the reward and transition functions. These results highlight the role of subteam structures in achieving near-optimal coordination. In our empirical study, we propose a heuristic for dynamically constructing context-aware DAGs that induce subteam policies, and demonstrate its effectiveness across tabular and deep MARL benchmarks.

### ACKNOWLEDGMENTS

This work was supported by AF Office of Scientific Research (AFOSR) under grant FA9550-25-1-0318 and with the generous support of the Amazon Research Award program. Qi Zhang acknowledges funding support from National Science Foundation (NSF) award 2544947 and NSF CAREER award 2544948.

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

# A   PROOFS

## A.1   THE SMOOTHNESS BOUND

**Lemma 4.** $L_\lambda(\theta)$ is $\beta_\lambda$-smooth with $\beta_\lambda = \frac{8N}{(1-\gamma)^3} + \frac{2\lambda N}{|\mathcal{S}|}$.

*Proof.* Lemma A.3 in Chen & Zhang (2023) establishes that $V_\theta$ is $\frac{8N}{(1-\gamma)^3}$-smooth. From the perspective of the augmented state, Lemma D.4 in Agarwal et al. (2021) implies that the regularizer for each agent $i$ is $\frac{2\lambda}{|\mathcal{S}||\mathcal{A}^{\mathcal{P}^i}|}$-smooth. Therefore, the overall smoothness of $L_\lambda(\theta)$ is bounded above by

$$\frac{8N}{(1-\gamma)^3} + \sum_i \frac{2\lambda}{|\mathcal{S}||\mathcal{A}^{\mathcal{P}^i}|}.$$

Since $\sum_i \frac{1}{|\mathcal{A}^{\mathcal{P}^i}|} \le N$, we have

$$\sum_i \frac{2\lambda}{|\mathcal{S}||\mathcal{A}^{\mathcal{P}^i}|} \le \frac{2\lambda N}{|\mathcal{S}|}.$$

Thus, $\beta_\lambda = \frac{8N}{(1-\gamma)^3} + \frac{2\lambda N}{|\mathcal{S}|}$ serves as an upper bound on the smoothness of $L_\lambda(\theta)$. $\square$

## A.2   PROOF OF LEMMA 1

*Proof.*

$$\frac{\partial L_\lambda(\theta)}{\partial \theta^i(s, a^{\mathcal{P}^i}, a^i)} = \underbrace{\frac{\partial V_\theta(\mu)}{\partial \theta^i(s, a^{\mathcal{P}^i}, a^i)}}_{(1)} - \underbrace{\frac{\partial \left( \lambda \sum_{i=1}^N \mathbb{E}_{s, a^{\mathcal{P}^i} \sim \mathrm{Unif}_{\mathcal{S} \times \mathcal{A}^{\mathcal{P}^i}}} \left[ \mathrm{KL}(\mathrm{Unif}_{\mathcal{A}^i}, \pi_\theta(\cdot|s, a^{\mathcal{P}^i})) \right] \right)}{\partial \theta^i(s, a^{\mathcal{P}^i}, a^i)}}_{(2)}$$

According to Lemma 5.1 in Chen & Zhang (2023),

$$(1) = \frac{1}{1-\gamma} d_\mu^{\pi_\theta}(s, a^{\mathcal{P}^i}) \pi_{\theta^i}^i(a^i|s, a^{\mathcal{P}^i}) A_\theta^i(s, a^{\mathcal{P}^i}, a^i).$$

By the definition of KL Divergence,

$$(2) = \frac{\partial \left( \lambda \sum_{i=1}^N \mathbb{E}_{s, a^{\mathcal{P}^i} \sim \mathrm{Unif}_{\mathcal{S} \times \mathcal{A}^{\mathcal{P}^i}}} \lambda \sum_{i=1}^N \left( \frac{\sum_{s, a^{\mathcal{P}^i}, a^i} \log \pi_{\theta^i}^i(a^i|s, a^{\mathcal{P}^i})}{|\mathcal{S}||\mathcal{A}^{\mathcal{P}^i}||\mathcal{A}^i|} + \log |\mathcal{A}^i| \right) \right)}{\partial \theta^i(s, a^{\mathcal{P}^i}, a^i)}$$

$$= -\frac{\lambda}{|\mathcal{S}||\mathcal{A}^{\mathcal{P}^i}|} \left( \frac{1}{|\mathcal{A}^i|} - \pi_{\theta^i}^i(a^i|s, a^{\mathcal{P}^i}) \right).$$

Therefore,

$$\frac{\partial L_\lambda(\theta)}{\partial \theta^i(s, a^{\mathcal{P}^i}, a^i)}$$
$$= (1) - (2)$$
$$= \frac{1}{1-\gamma} d_\mu^{\pi_\theta}(s, a^{\mathcal{P}^i}) \pi_{\theta^i}^i(a^i|s, a^{\mathcal{P}^i}) A_\theta^i(s, a^{\mathcal{P}^i}, a^i) + \frac{\lambda}{|\mathcal{S}||\mathcal{A}^{\mathcal{P}^i}|} \left( \frac{1}{|\mathcal{A}^i|} - \pi_{\theta^i}^i(a^i|s, a^{\mathcal{P}^i}) \right).$$

$\square$

### A.3    PROOF OF LEMMA 2

*Proof.* The proof extends the proof of Theorem 5.2 in Agarwal et al. (2021) by the usage of the multi-agent performance difference lemma (Lemma C.1 in Leonardos et al. (2021)).

Similar to the proof of Theorem 5.2 in Agarwal et al. (2021), we can establish an upper bound on the advantage function $A_\theta^i(s, a^{\mathcal{P}^i}, a^i)$ for any $(s, a^{\mathcal{P}^i}, a^i)$-pair. It suffices to consider the case where $A_\theta^i(s, a^{\mathcal{P}^i}, a^i) \geq 0$ (since when $A_\theta^i(s, a^{\mathcal{P}^i}, a^i) < 0$, any positive number serves as a valid upper bound):

$$
\begin{aligned}
&\lambda/(2|\mathcal{S}||\mathcal{A}| \max_j |\mathcal{A}^j|) \\
\geq& \lambda/(2|\mathcal{S}||\mathcal{A}^{\mathcal{P}^i}||\mathcal{A}^i|) \qquad (=: \epsilon_{\mathrm{opt}}) \\
\geq& \frac{\partial L_\lambda(\theta)}{\partial \theta^i(s, a^{\mathcal{P}^i}, a^i)} \\
\overset{(i)}{=}& \frac{1}{1-\gamma} d_\mu^{\pi_\theta}(s, a^{\mathcal{P}^i}) \pi_{\theta^i}^i(a^i|s, a^{\mathcal{P}^i}) A_\theta^i(s, a^{\mathcal{P}^i}, a^i) + \frac{\lambda}{|\mathcal{S}||\mathcal{A}^{\mathcal{P}^i}|} \left( \frac{1}{|\mathcal{A}^i|} - \pi_{\theta^i}^i(a^i|s, a^{\mathcal{P}^i}) \right) \\
\geq& \frac{\lambda}{|\mathcal{S}||\mathcal{A}^{\mathcal{P}^i}|} \left( \frac{1}{|\mathcal{A}^i|} - \pi_{\theta^i}^i(a^i|s, a^{\mathcal{P}^i}) \right)
\end{aligned}
$$

where the last inequality is due to $A_\theta^i(s, a^{\mathcal{P}^i}, a^i) \geq 0$, and by rearranging we get

$$
\pi_{\theta^i}^i(a^i|s, a^{\mathcal{P}^i}) \geq \frac{1}{2|\mathcal{A}^i|}. \tag{9}
$$

Solving (i) for $A_\theta^i(s, a^{\mathcal{P}^i}, a^i)$, we have

$$
\begin{aligned}
A_\theta^i(s, a^{\mathcal{P}^i}, a^i) =& \frac{1-\gamma}{d_\mu^{\pi_\theta}(s, a^{\mathcal{P}^i})} \left( \frac{1}{\pi_{\theta^i}^i(a^i|s, a^{\mathcal{P}^i})} \frac{\partial L_\lambda(\theta)}{\partial \theta^i(s, a^{\mathcal{P}^i}, a^i)} + \frac{\lambda}{|\mathcal{S}||\mathcal{A}^{\mathcal{P}^i}|} \left( 1 - \frac{1}{\pi_{\theta^i}^i(a^i|s, a^{\mathcal{P}^i})|\mathcal{A}^i|} \right) \right) \\
\leq& \frac{1-\gamma}{d_\mu^{\pi_\theta}(s, a^{\mathcal{P}^i})} \left( 2|\mathcal{A}^i|\epsilon_{\mathrm{opt}} + \frac{\lambda}{|\mathcal{S}||\mathcal{A}^{\mathcal{P}^i}|} \right) \qquad (\pi_{\theta^i}^i(a^i|s, a^{\mathcal{P}^i}) \geq \frac{1}{2|\mathcal{A}^i|}) \\
\leq& \frac{2(1-\gamma)\lambda}{d_\mu^{\pi_\theta}(s, a^{\mathcal{P}^i})|\mathcal{S}||\mathcal{A}^{\mathcal{P}^i}|} \qquad (\epsilon_{\mathrm{opt}} = \lambda/(2|\mathcal{S}||\mathcal{A}^{\mathcal{P}^i}||\mathcal{A}^i|))
\end{aligned} \tag{10}
$$

We are now ready to use the multi-agent performance difference lemma on two BN policies with only agent $i$'s parameters changed. For convenience, denote $\sum_{a^{-\mathcal{P}^i}} \pi_\theta(a^{-\mathcal{P}^i}, a^{\mathcal{P}^i}|s)$ as $\bar{\pi}_\theta^{\mathcal{P}^i}(\cdot|s)$ so that $d_\mu^{\pi_\theta}(s, a^{\mathcal{P}^i}) = d_\mu^{\pi_\theta}(s)\bar{\pi}_\theta^{\mathcal{P}^i}(\cdot|s)$. Fix an arbitrary agent $i \in \mathcal{N}$ and suppose it deviates from $\pi_{\theta^i}^i$ to an optimal policy $\pi_{\tilde{\theta}^i}^i$ w.r.t. the corresponding single-agent MDP specified by $\theta^{-i}$. Let $\theta' = [\theta^{-i}, \tilde{\theta}^i]$ be the parameters of any joint policy where only agent $i$'s parameters are changed to the optimal

policy in the the corresponding single-agent MDP. We have

$$V_{\theta'}(\mu) - V_\theta(\mu)$$

$$= \frac{1}{1-\gamma} \mathbb{E}_{\bar{s} \sim d_\mu^{\pi_{\theta'}}} \mathbb{E}_{\bar{a} \sim \pi_{\theta'}} \left[ A_\theta(\bar{s}, \bar{a}) \right] \quad \text{(performance difference lemma)}$$

$$= \frac{1}{1-\gamma} \mathbb{E}_{\bar{s} \sim d_\mu^{\pi_{\theta'}}(\cdot)} \mathbb{E}_{\bar{a}^{\mathcal{P}^i} \sim \bar{\pi}_{\theta'}^{\mathcal{P}^i}(\cdot|\bar{s})} \mathbb{E}_{\bar{a}^i \sim \pi_{\hat{\theta}^i}^i(\cdot|\bar{s}, \bar{a}^{\mathcal{P}^i})} \mathbb{E}_{\bar{a}^{-\mathcal{P}^i}_+ \sim \pi_{\theta'}^{-\mathcal{P}^i_+}(\cdot|\bar{s}, a^{\mathcal{P}^i}_+)} \left[ Q_\theta(\bar{s}, \bar{a}^{\mathcal{P}^i}, \bar{a}^i, \bar{a}^{-\mathcal{P}^i}_+) - V_\theta(\bar{s}) \right]$$

$$\left( \text{ Since } (\theta')^{-i} = \theta^{-i} \text{ which means } \bar{\pi}_{\theta'}^{\mathcal{P}^i}(\cdot|\bar{s}) = \bar{\pi}_\theta^{\mathcal{P}^i}(\cdot|\bar{s}), \pi_{\theta'}^{-\mathcal{P}^i_+}(\cdot|\bar{s}, a^{\mathcal{P}^i}_+) = \pi_\theta^{-\mathcal{P}^i_+}(\cdot|\bar{s}, a^{\mathcal{P}^i}_+) \right)$$

$$= \frac{1}{1-\gamma} \mathbb{E}_{\bar{s} \sim d_\mu^{\pi_{\theta'}}(\cdot)} \mathbb{E}_{\bar{a}^{\mathcal{P}^i} \sim \bar{\pi}_\theta^{\mathcal{P}^i}(\cdot|\bar{s})} \mathbb{E}_{\bar{a}^i \sim \pi_{\hat{\theta}^i}^i(\cdot|\bar{s}, \bar{a}^{\mathcal{P}^i})} \mathbb{E}_{\bar{a}^{-\mathcal{P}^i}_+ \sim \pi_\theta^{-\mathcal{P}^i_+}(\cdot|\bar{s}, a^{\mathcal{P}^i}_+)} \left[ Q_\theta(\bar{s}, \bar{a}^{\mathcal{P}^i}, \bar{a}^i, \bar{a}^{-\mathcal{P}^i}_+) - V_\theta(\bar{s}) \right]$$

$$= \frac{1}{1-\gamma} \mathbb{E}_{\bar{s} \sim d_\mu^{\pi_{\theta'}}(\cdot)} \mathbb{E}_{\bar{a}^{\mathcal{P}^i} \sim \bar{\pi}_\theta^{\mathcal{P}^i}(\cdot|\bar{s})} \mathbb{E}_{\bar{a}^i \sim \pi_{\hat{\theta}^i}^i(\cdot|\bar{s}, \bar{a}^{\mathcal{P}^i})} \left[ Q_\theta^i(\bar{s}, \bar{a}^{\mathcal{P}^i}, \bar{a}^i) - V_\theta(\bar{s}) \right]$$

$$= \frac{1}{1-\gamma} \mathbb{E}_{\bar{s} \sim d_\mu^{\pi_{\theta'}}(\cdot)} \mathbb{E}_{\bar{a}^{\mathcal{P}^i} \sim \bar{\pi}_\theta^{\mathcal{P}^i}(\cdot|\bar{s})} \mathbb{E}_{\bar{a}^i \sim \pi_{\hat{\theta}^i}^i(\cdot|\bar{s}, \bar{a}^{\mathcal{P}^i})} \left[ Q_\theta^i(\bar{s}, \bar{a}^{\mathcal{P}^i}, \bar{a}^i) - V_\theta(\bar{s}) \right]$$

$$= \frac{1}{1-\gamma} \mathbb{E}_{\bar{s} \sim d_\mu^{\pi_{\theta'}}(\cdot)} \mathbb{E}_{\bar{a}^{\mathcal{P}^i} \sim \bar{\pi}_\theta^{\mathcal{P}^i}(\cdot|\bar{s})} \mathbb{E}_{\bar{a}^i \sim \pi_{\hat{\theta}^i}^i(\cdot|\bar{s}, \bar{a}^{\mathcal{P}^i})} \left[ A_\theta^i(\bar{s}, \bar{a}^{\mathcal{P}^i}, \bar{a}^i) + Q_\theta^i(\bar{s}, \bar{a}^{\mathcal{P}^i}) - V_\theta(\bar{s}) \right]$$

$$= \frac{1}{1-\gamma} \mathbb{E}_{\bar{s} \sim d_\mu^{\pi_{\theta'}}(\cdot)} \mathbb{E}_{\bar{a}^{\mathcal{P}^i} \sim \bar{\pi}_\theta^{\mathcal{P}^i}(\cdot|\bar{s})} \mathbb{E}_{\bar{a}^i \sim \pi_{\hat{\theta}^i}^i(\cdot|\bar{s}, \bar{a}^{\mathcal{P}^i})} A_\theta^i(\bar{s}, \bar{a}^{\mathcal{P}^i}, \bar{a}^i)$$

$$+ \frac{1}{1-\gamma} \mathbb{E}_{\bar{s} \sim d_\mu^{\pi_{\theta'}}(\cdot)} \mathbb{E}_{\bar{a}^{\mathcal{P}^i} \sim \bar{\pi}_\theta^{\mathcal{P}^i}(\cdot|\bar{s})} \mathbb{E}_{\bar{a}^i \sim \pi_{\hat{\theta}^i}^i(\cdot|\bar{s}, \bar{a}^{\mathcal{P}^i})} \left[ Q_\theta^i(\bar{s}, \bar{a}^{\mathcal{P}^i}) - V_\theta(\bar{s}) \right]$$

$$\leq \frac{1}{1-\gamma} \mathbb{E}_{\bar{s} \sim d_\mu^{\pi_{\theta'}}(\cdot)} \mathbb{E}_{\bar{a}^{\mathcal{P}^i} \sim \bar{\pi}_\theta^{\mathcal{P}^i}(\cdot|\bar{s})} \mathbb{E}_{\bar{a}^i \sim \pi_{\hat{\theta}^i}^i(\cdot|\bar{s}, \bar{a}^{\mathcal{P}^i})} \frac{2(1-\lambda)\lambda}{d_\mu^{\pi_\theta}(\bar{s}, \bar{a}^{\mathcal{P}^i})|\mathcal{S}||\mathcal{A}^{\mathcal{P}^i}|}$$

$$+ \frac{1}{1-\gamma} \mathbb{E}_{\bar{s} \sim d_\mu^{\pi_{\theta'}}(\cdot)} \mathbb{E}_{\bar{a}^{\mathcal{P}^i} \sim \bar{\pi}_\theta^{\mathcal{P}^i}(\cdot|\bar{s})} \left[ Q_\theta^i(\bar{s}, \bar{a}^{\mathcal{P}^i}) - V_\theta(\bar{s}) \right]$$

$$= \frac{1}{1-\gamma} \mathbb{E}_{\bar{s} \sim d_\mu^{\pi_{\theta'}}(\cdot)} \mathbb{E}_{\bar{a}^{\mathcal{P}^i} \sim \bar{\pi}_\theta^{\mathcal{P}^i}(\cdot|\bar{s})} \frac{2(1-\gamma)\lambda}{d_\mu^{\pi_\theta}(\bar{s}, \bar{a}^{\mathcal{P}^i})|\mathcal{S}||\mathcal{A}^{\mathcal{P}^i}|}$$

$$= \frac{1}{1-\gamma} \mathbb{E}_{\bar{s} \sim d_\mu^{\pi_{\theta'}}(\cdot)} \mathbb{E}_{\bar{a}^{\mathcal{P}^i} \sim \bar{\pi}_\theta^{\mathcal{P}^i}(\cdot|\bar{s})} \frac{2(1-\gamma)\lambda}{d_\mu^{\pi_\theta}(s)\bar{\pi}_\theta^{\mathcal{P}^i}(\cdot|s)|\mathcal{S}||\mathcal{A}^{\mathcal{P}^i}|}$$

$$= \mathbb{E}_{\bar{s} \sim d_\mu^{\pi_{\theta'}}(\cdot)} \frac{2\lambda}{d_\mu^{\pi_\theta}(\bar{s})|\mathcal{S}||\mathcal{A}^{\mathcal{P}^i}|}$$

$$= \sum_{\bar{s}} d_\mu^{\pi_{\theta'}}(\bar{s}) \frac{2\lambda}{d_\mu^{\pi_\theta}(\bar{s})|\mathcal{S}||\mathcal{A}^{\mathcal{P}^i}|}$$

$$\leq \frac{2\lambda}{|\mathcal{A}^{\mathcal{P}^i}|} \max_s \left( \frac{d_\mu^{\pi_{\theta'}}(s)}{d_\mu^{\pi_\theta}(s)} \right) \leq \frac{2\lambda}{|\mathcal{A}^{\mathcal{P}^i}|} M \leq 2\lambda M.$$

By definition of $\epsilon$-approximate equilibrium, we know that the BN (joint) policy $\pi_\theta = (\pi_{\theta^1}^1, ..., \pi_{\theta^N}^N)$ is a $2\lambda M$-approximate equilibrium. $\square$

### A.4 PROOF OF THEOREM 1

*Proof.* Since $L_\lambda(\theta)$ is $\beta_\lambda$-smooth, we have

$$\min_{t \leq T} \left\| \nabla_\theta L_\lambda(\theta^{(t)}) \right\|_2^2 \leq \frac{2\beta_\lambda(L_\lambda(\theta^*) - L_\lambda(\theta_0))}{T} \leq \frac{2\beta_\lambda(V_{\max} - V_{\min})}{T} \leq \frac{2\beta_\lambda}{T(1-\gamma)}$$

where the second inequality holds because we initialize $\theta_0 = 0$. We can choose $T$ large enough such that

$$\sqrt{\frac{2\beta_\lambda}{T(1-\gamma)}} \leq \lambda/(2|\mathcal{S}| \max_i |\mathcal{A}^i|).$$

Solving the above inequality we obtain $T \geq \frac{8\beta_\lambda|\mathcal{S}|^2 \max_i |\mathcal{A}^i|^2}{\lambda^2(1-\gamma)}$. By Lemma 2, we should set $\lambda = \frac{\epsilon}{2M}$ to achieve the specified equilibrium-gap of $\epsilon$. Plugging in $\lambda = \frac{\epsilon}{2M}$ and $\beta_\lambda = \frac{8N}{(1-\gamma)^3} + \frac{2\lambda N}{|\mathcal{S}|}$, we

have

$$T \geq \frac{32M^2|\mathcal{S}|^2 \max_i |\mathcal{A}^i|^2 \beta_\lambda}{\epsilon^2(1-\gamma)}$$

$$= \frac{256NM^2|\mathcal{S}|^2 \max_i |\mathcal{A}^i|^2}{(1-\gamma)^4\epsilon^2} + \frac{64\lambda NM^2|\mathcal{S}| \max_i |\mathcal{A}^i|^2}{(1-\gamma)\epsilon^2}$$

$$= \frac{256NM^2|\mathcal{S}|^2 \max_i |\mathcal{A}^i|^2}{(1-\gamma)^4\epsilon^2} + \frac{32NM|\mathcal{S}| \max_i |\mathcal{A}^i|^2}{(1-\gamma)\epsilon}$$

$\square$

**Lemma 5** (Properties of subteams-decomposed cooperative MGs). *Suppose the underlying DAG of a BN policy $\pi$ satisfies all conditions in Assumption 3 with partition $\{\mathcal{C}_k\}_{1 \leq k \leq K}$ decomposing the cooperative MG of interest with errors $(\epsilon_P, \epsilon_r)$. We have*

*(i) **Factorized joint policy.** For any state $s \in \mathcal{S}$ and joint action $a \in \mathcal{A}$, the BN policy is factorized into the $K$ subteams as: $\pi(a|s) = \prod_{k=1}^{K} \pi^{\mathcal{C}_k}(a^{\mathcal{C}_k}|s)$.*

*(ii) **Decomposed value functions.** The global action-value function $Q_\pi(s,a)$ and state-value function $V_\pi(s)$ are decomposed additively by the subteams as*

$$Q_\pi(s,a) = \sum_{k=1}^{K} Q_\pi^k(s, a^{\mathcal{C}_k}) + \epsilon_{Q_\pi}(s,a) \quad and \quad V_\pi(s) = \sum_{k=1}^{K} V_\pi^k(s) + \epsilon_{V_\pi}(s)$$

*where* $\quad Q_\pi^k(s, a^{\mathcal{C}_k}) := r^k(s, a^{\mathcal{C}_k}) + \gamma \mathbb{E}_{s' \sim P^k(\cdot|s, a^{\mathcal{C}_k})} [V_\pi(s')],$

$$V_\pi^k(s) := \mathbb{E}_{a^{\mathcal{C}_k} \sim \pi^{\mathcal{C}_k}(\cdot|s)} [Q_\pi^k(s, a^{\mathcal{C}_k})],$$

$$\epsilon_{Q_\pi}(s,a) := \epsilon_r(s,a) + \gamma \mathbb{E}_{s' \sim \epsilon_P(\cdot|s,a)} [V_\pi(s')], \quad \epsilon_{V_\pi}(s) := \mathbb{E}_{a \sim \pi(\cdot|s)} [\epsilon_{Q_\pi}(s,a)]$$

*(iii) **Marginal Consistency Property.** This property captures that each subteam's partially aggregated Q-function (i.e., after marginalizing out actions of agents outside the subteam) differs from the value function $V$ in exactly the same way that the subteam's local $\widetilde{Q}$ differs from its local baseline $\widetilde{V}$. Formally, for every subteam $C^k$ and any joint action $a^{\mathcal{C}_k}$ of agents in that subteam,*

$$Q_\pi(s, a^{\mathcal{C}_k}) - V_\pi(s) = Q_\pi^k(s, a^{\mathcal{C}_k}) - V_\pi^k(s) + \left(\epsilon_{Q_\pi}(s, a^{\mathcal{C}_k}) - \epsilon_{V_\pi}(s)\right).$$

*where*

$$\epsilon_{Q_\pi}(s, a^{\mathcal{C}_k}) := \mathbb{E}_{a^{-\mathcal{C}_k} \sim \pi^{-\mathcal{C}_k}(\cdot|s)} \left[\epsilon_Q(s, a^{\mathcal{C}_k}, a^{-\mathcal{C}_k})\right]$$

*Proof.* **(i) Factorized Joint Policy Functions.** No edges exist between agents in different subteams, so each subteam $\mathcal{C}_k$'s local policy depends only on $s$ and its intra-subteam parents:

$$\pi^{\mathcal{C}_k}(a^{\mathcal{C}_k} \mid s) = \prod_{i \in \mathcal{C}_k} \pi^i(a^i \mid s, a^{\mathcal{P}^i}), \quad \text{where } \mathcal{P}^i \subseteq \mathcal{C}_k.$$

Because these subteams are disjoint, the overall joint policy factors:

$$\pi(a \mid s) = \prod_{k=1}^{K} \pi^{\mathcal{C}_k}(a^{\mathcal{C}_k} \mid s).$$

Hence Property (i) follows.

**(ii) Factorized Critic and Value Function.** We first show $Q(s,a)$ factorizes. By definition,

$$Q(s,a) = r(s,a) + \gamma \mathbb{E}_{s' \sim P(\cdot|s,a)} [V_\pi(s')].$$

By the factorized transition and reward defined in Equation (5), we have

$$Q(s,a) = \epsilon_r(s,a) + \sum_{k=1}^{K} r^k(s, a^{\mathcal{C}_k}) + \gamma \left( \mathbb{E}_{s' \sim \epsilon_P(s'|s,a)} [V_\pi(s')] + \sum_{k=1}^{K} \mathbb{E}_{s' \sim P^k(\cdot|s, a^{\mathcal{C}_k})} [V_\pi(s')] \right).$$

Define

$$Q_\pi^k(s, a^{\mathcal{C}_k}) := r^k(s, a^{\mathcal{C}_k}) + \gamma \mathbb{E}_{s' \sim P^k(\cdot|s, a^{\mathcal{C}_k})}\big[V_\pi(s')\big].$$

and

$$\epsilon_{Q_\pi}(s, a) := \epsilon_r(s, a) + \gamma \mathbb{E}_{s' \sim \epsilon_P(\cdot|s, a)}\big[V_\pi(s')\big],$$

Hence

$$Q(s, a) = \sum_{k=1}^{K} Q_\pi^k(s, a^{\mathcal{C}_k}) + \epsilon_{Q_\pi}(s, a).$$

We can then show $V_\pi(s)$ factorizes as the following

$$V_\pi(s) = \mathbb{E}_{a \sim \pi(\cdot|s)}\big[Q(s, a)\big]$$
$$= \mathbb{E}_{a \sim \pi(\cdot|s)}\Big[\sum_{k=1}^{K} Q_\pi^k(s, a^{\mathcal{C}_k}) + \epsilon_{Q_\pi}(s, a)\Big]$$

Define

$$V_\pi^k(s) := \mathbb{E}_{a^{\mathcal{C}_k} \sim \pi^{\mathcal{C}_k}(\cdot \mid s)}\Big[Q_\pi^k(s, a^{\mathcal{C}_k})\Big],$$

and

$$\epsilon_{V_\pi}(s) := \mathbb{E}_{a \sim \pi(\cdot \mid s)}\Big[\epsilon_{Q_\pi}(s, a)\Big]$$

Hence,

$$V_\pi(s) = \sum_{k=1}^{K} V_\pi^k(s) + \epsilon_{V_\pi}(s),$$

Therefore, both $Q(s, a)$ and $V_\pi(s)$ factor over the subteams, establishing Property (ii).

**(iii) Marginal Consistency Property.** For each subteam $\mathcal{C}_k$, we first define

$$\epsilon_{Q_\pi}(s, a^{\mathcal{C}_k}) := \mathbb{E}_{a^{-\mathcal{C}_k} \sim \pi^{-\mathcal{C}_k}(\cdot|s)}\Big[\epsilon_Q(s, a^{\mathcal{C}_k}, a^{-\mathcal{C}_k})\Big]$$

By definition we have

$$Q_\pi(s, a^{\mathcal{C}_k}) = \mathbb{E}_{a^{-\mathcal{C}_k} \sim \pi^{-\mathcal{C}_k}(\cdot|s)}\Big[\sum_{\ell=1}^{K} Q_\pi^\ell(s, a^{\mathcal{C}_\ell}) + \epsilon_Q(s, a^{\mathcal{C}_k}, a^{-\mathcal{C}_k})\Big] + \epsilon_{Q_\pi}(s, a^{\mathcal{C}_k})$$
$$= Q_\pi^k(s, a^{\mathcal{C}_k}) + \sum_{\ell \neq k} \mathbb{E}_{a^{-\mathcal{C}_k} \sim \pi^{-\mathcal{C}_k}(\cdot|s)}\big[Q_\pi^\ell(s, a^{\mathcal{C}_\ell})\big] + \epsilon_{Q_\pi}(s, a^{\mathcal{C}_k})$$
$$= Q_\pi^k(s, a^{\mathcal{C}_k}) + \sum_{\ell \neq k} \mathbb{E}_{a^{-(\mathcal{C}_k \cup \mathcal{C}_\ell)} \sim \pi^{-(\mathcal{C}_k \cup \mathcal{C}_\ell)}(\cdot|s)}\Big[\mathbb{E}_{a^{\mathcal{C}_\ell} \sim \pi}\big[Q_\pi^\ell(s, a^{\mathcal{C}_\ell})\big]\Big] + \epsilon_{Q_\pi}(s, a^{\mathcal{C}_k})$$
$$= Q_\pi^k(s, a^{\mathcal{C}_k}) + \sum_{\ell \neq k} \mathbb{E}_{a^{-(\mathcal{C}_k \cup \mathcal{C}_\ell)} \sim \pi}\big[V^\ell(s)\big] + \epsilon_{Q_\pi}(s, a^{\mathcal{C}_k})$$
$$= Q_\pi^k(s, a^{\mathcal{C}_k}) + \sum_{\ell \neq k} V^\ell(s) + \epsilon_{Q_\pi}(s, a^{\mathcal{C}_k})$$

and

$$V_\pi(s) = \sum_{\ell=1}^{K} V_\pi^k(s) + \epsilon_{V_\pi}(s) = V_\pi^k(s) + \sum_{\ell \neq k} V^\ell(s) + \epsilon_{V_\pi}(s).$$

Hence,

$$Q_\pi(s, a^{\mathcal{C}_k}) - V_\pi(s) = Q_\pi^k(s, a^{\mathcal{C}_k}) - V_\pi^k(s) + \big(\epsilon_{Q_\pi}(s, a^{\mathcal{C}_k}) - \epsilon_{V_\pi}(s)\big).$$

This establishes the Marginal Consistency Property (iii) and completes the proof of Lemma 5. $\square$

Below we define the bounds for the decomposition errors for the transition and reward.

**Lemma 6** (Bound on Deviations in Factorized Critic and Value Functions). *For any policy $\pi$, state $s \in \mathcal{S}$, joint action $a \in \mathcal{A}$, and subteam action $a^{\mathcal{C}_k}$, the following deviations are upper bounded:*

$$\left\{ |\epsilon_{Q_\pi}(s,a)|, \ |\epsilon_{V_\pi}(s)|, \ |\epsilon_{Q_\pi}(s,a^{\mathcal{C}_k})| \right\} \leq \underbrace{|\epsilon_r|}_{\text{reward error}} + \underbrace{|S||\epsilon_P|}_{\text{transition error}} \cdot \underbrace{\gamma/(1-\gamma)}_{\text{cumulative reward bound}}$$

The bound consists of two parts: 1) $\epsilon_r$ quantifies the one-step error due to reward decomposition, and 2) the second term captures the cumulative effect of transition decomposition, scaled by the worst-case return bound $\gamma/(1-\gamma)$. The overall deviation becomes small when both decomposition errors are small, and vanishes entirely when $K = 1$, in which case no decomposition is needed.

*Proof.* We begin with the definition:

$$\epsilon_{Q_\pi}(s,a) := \epsilon_r(s,a) + \gamma \mathbb{E}_{s' \sim \Delta P(\cdot|s,a)}[V_\pi(s')].$$

Taking the absolute value and applying the triangle inequality:

$$|\epsilon_{Q_\pi}(s,a)| \leq |\epsilon_r(s,a)| + \gamma \sum_{s'} |\epsilon_P(s' \mid s,a)| \cdot |V_\pi(s')|.$$

Since we have:

$$|\epsilon_r(s,a)| \leq |\epsilon_r|, \quad \sum_{s'} |\epsilon_P(s' \mid s,a)| \leq |S||\epsilon_P|, \quad |V_\pi(s')| \leq \frac{1}{1-\gamma},$$

we obtain:

$$|\epsilon_{Q_\pi}(s,a)| \leq |\epsilon_r| + \gamma |S||\epsilon_P| \cdot \frac{1}{1-\gamma} = |\epsilon_r| + |\epsilon_P| \frac{\gamma |S|}{1-\gamma}.$$

Now consider the deviation in the value function:

$$\epsilon_{V_\pi}(s) := \mathbb{E}_{a \sim \pi(\cdot|s)}[\epsilon_{Q_\pi}(s,a)].$$

Applying Jensen's inequality:

$$|\epsilon_{V_\pi}(s)| \leq \mathbb{E}_a \left[|\epsilon_{Q_\pi}(s,a)|\right] \leq |\epsilon_r| + |\epsilon_P| \frac{\gamma |S|}{1-\gamma}.$$

Finally, for any subteam $\mathcal{C}_k$:

$$\epsilon_{Q_\pi}(s, a^{\mathcal{C}_k}) := \mathbb{E}_{a^{-\mathcal{C}_k} \sim \pi^{-\mathcal{C}_k}(\cdot|s)}[\epsilon_Q(s, a^{\mathcal{C}_k}, a^{-\mathcal{C}_k})].$$

Again applying Jensen's inequality:

$$|\epsilon_{Q_\pi}(s, a^{\mathcal{C}_k})| \leq \mathbb{E}_{a^{-\mathcal{C}_k}} \left[|\epsilon_Q(s, a^{\mathcal{C}_k}, a^{-\mathcal{C}_k})|\right] \leq |\epsilon_r| + |\epsilon_P| \frac{\gamma |S|}{1-\gamma}.$$

$\square$

## A.5 Proof of Lemma 3

*Proof.* By bound on the advantage inequality (10), we know that $\forall s, \mathcal{A}^{\mathcal{P}^i}, a^i$,

$$A_\theta^i(s, a^{\mathcal{P}^i}, a^i) \leq \frac{2(1-\gamma)\lambda}{d_\mu^{\pi_\theta}(s, a^{\mathcal{P}^i})|\mathcal{S}||\mathcal{A}^{\mathcal{P}^i}|} = \frac{2(1-\gamma)\lambda}{d_\mu^{\pi_\theta}(s)\pi_\theta(a^{\mathcal{P}^i}|s)|\mathcal{S}||\mathcal{A}^{\mathcal{P}^i}|}. \tag{11}$$

By inequality (9), we know

$$\pi_\theta(a^{\mathcal{P}^i}|s) = \prod_{j \in a^{\mathcal{P}^i}} \pi_\theta(a^j|s, a^{\mathcal{P}^j}) \geq \prod_{j \in a^{\mathcal{P}^i}} \frac{1}{2|\mathcal{A}^j|} = \frac{1}{2^{|a^{\mathcal{P}^i}|}} \frac{1}{|\mathcal{A}^{\mathcal{P}^i}|}.$$

Plugging in (11), we have

$$A_\theta^i(s, a^{\mathcal{P}^i}, a^i) \le \frac{2^{|a^{\mathcal{P}^i}|+1}(1-\gamma)\lambda}{d_\mu^{\pi_\theta}(s)|\mathcal{S}|}.$$

Assume without loss of generality that agents in $\mathcal{C}_k$ have agent id $1, 2 \cdots |\mathcal{C}_k|$ and have a corresponding topological ordering of $1, 2 \cdots \cdots |\mathcal{C}_k|$ determined by $G$. Note that in this case, since each subteam $\mathcal{C}_k$ is disjoint to other subteams and agents within each subteam are fully connected, we have $\forall i \in \mathcal{C}_k, a^{\mathcal{P}^i} = [a^{\mathcal{P}_+^{i-1}}]$, which means that

$$
\begin{aligned}
Q_\theta(s, a^{\mathcal{P}^i}) &= \mathbb{E}_{\bar{a}^{-\mathcal{P}^i} \sim \pi_\theta(\cdot|s, a^{\mathcal{P}^i})} \left[ Q_\theta(s, a^{\mathcal{P}^i}, \bar{a}^{-\mathcal{P}^i}) \right] \\
&= \mathbb{E}_{\bar{a}^{-\mathcal{P}_+^{i-1}} \sim \pi_\theta(\cdot|s, a^{\mathcal{P}_+^{i-1}})} \left[ Q_\theta(s, a^{\mathcal{P}_+^{i-1}}, \bar{a}^{-\mathcal{P}_+^{i-1}}) \right] = Q_\theta(s, a^{\mathcal{P}_+^{i-1}}).
\end{aligned}
\tag{12}
$$

Following the reverse topological ordering, we have $\forall a^{\mathcal{C}_k} = [a^{\mathcal{P}^{|\mathcal{C}_k|}}, a^{|\mathcal{C}_k|}]$,

$$
\begin{aligned}
&Q_\theta(s, a^{\mathcal{C}_k}) \\
=&Q_\theta(s, a^{\mathcal{P}^{|\mathcal{C}_k|}}, a^{|\mathcal{C}_k|}) \\
\le&Q_\theta(s, a^{\mathcal{P}^{|\mathcal{C}_k|}}) + \frac{2^{(|\mathcal{C}_k|-1)+1}(1-\gamma)\lambda}{d_\mu^{\pi_\theta}(s)|\mathcal{S}|} \quad\quad \text{(by inequality (10))} \\
\le&Q_\theta(s, a^{\mathcal{P}^{|\mathcal{C}_k|-1}}) + \frac{(2^{|\mathcal{C}_k|} + 2^{|\mathcal{C}_k|-1})(1-\gamma)\lambda}{d_\mu^{\pi_\theta}(s)|\mathcal{S}|} \quad\quad \text{(by Equation (12))} \\
\le&Q_\theta(s, a^{\mathcal{P}^{|\mathcal{C}_k|-1}}) + \frac{(2^{|\mathcal{C}_k|} + 2^{|\mathcal{C}_k|-1})(1-\gamma)\lambda}{d_\mu^{\pi_\theta}(s)|\mathcal{S}|} \quad\quad \text{(by inequality (10))} \\
=&Q_\theta(s, a^{\mathcal{P}^{|\mathcal{C}_k|-2}}, a^{|\mathcal{C}_k|-2}) + \frac{(2^{|\mathcal{C}_k|} + 2^{|\mathcal{C}_k|-1})(1-\gamma)\lambda}{d_\mu^{\pi_\theta}(s)|\mathcal{S}|} \quad\quad \text{(by Equation (12))} \\
&\text{(keep doing the same procedure above)} \\
\le&Q_\theta(s, a^{\mathcal{P}^1}) + \frac{(\sum_{j=1}^{|\mathcal{C}_k|} 2^j)(1-\gamma)\lambda}{d_\mu^{\pi_\theta}(s)|\mathcal{S}|} \\
=&Q_\theta(s, a^{\mathcal{P}^1}) + \frac{(2^{|\mathcal{C}_k|+1} - 2)(1-\gamma)\lambda}{d_\mu^{\pi_\theta}(s)|\mathcal{S}|} \\
=&V_\theta(s) + \frac{(2^{|\mathcal{C}_k|+1} - 2)(1-\gamma)\lambda}{d_\mu^{\pi_\theta}(s)|\mathcal{S}|} \quad\quad \text{(since } a^{\mathcal{P}^1} = \emptyset\text{).}
\end{aligned}
$$

By property (iii) in Lemma 5, we get

$$Q_\theta^k(s, a^{\mathcal{C}_k}) \le V_\theta^k(s) + \frac{(2^{|\mathcal{C}_k|+1} - 2)(1-\gamma)\lambda}{d_\mu^{\pi_\theta}(s)|\mathcal{S}|} + \left(\epsilon_{V_\pi}(s) - \epsilon_{Q_\pi}(s, a^{\mathcal{C}_k})\right).$$

By property (ii), we can bound the difference between Global $Q$ and $V$ by the following:

$$Q_\theta(s,a) = \sum_{k=1}^{K} Q_\theta^k(s, a^{\mathcal{C}_k}) + \epsilon_Q(s,a)$$

$$\leq \sum_{k=1}^{K} \left( V_\theta^k(s) + \frac{(2^{|\mathcal{C}_k|+1} - 2)(1-\gamma)\lambda}{d_\mu^{\pi_\theta}(s)|\mathcal{S}|} + \left( \epsilon_{V_\pi}(s) - \epsilon_{Q_\pi}(s, a^{\mathcal{C}_k}) \right) \right) + \epsilon_Q(s,a)$$

$$= \sum_{k=1}^{K} V_\theta^k(s) + \frac{(\sum_{k=1}^{K} 2^{|\mathcal{C}_k|+1} - 2K)(1-\gamma)\lambda}{d_\mu^{\pi_\theta}(s)|\mathcal{S}|} + \sum_{k=1}^{K} \left( \epsilon_{V_\pi}(s) - \epsilon_{Q_\pi}(s, a^{\mathcal{C}_k}) \right) + \epsilon_Q(s,a)$$

$$= V_\theta(s) + \frac{(\sum_{k=1}^{K} 2^{|\mathcal{C}_k|+1} - 2K)(1-\gamma)\lambda}{d_\mu^{\pi_\theta}(s)|\mathcal{S}|} + \sum_{k=1}^{K} \left( \epsilon_{V_\pi}(s) - \epsilon_{Q_\pi}(s, a^{\mathcal{C}_k}) \right) + \left( \epsilon_Q(s,a) - \epsilon_{V_\pi}(s) \right)$$

$$= V_\theta(s) + \frac{(\sum_{k=1}^{K} 2^{|\mathcal{C}_k|+1} - 2K)(1-\gamma)\lambda}{d_\mu^{\pi_\theta}(s)|\mathcal{S}|} + (k-1)\epsilon_{V_\pi}(s) - \sum_{k=1}^{K} \epsilon_{Q_\pi}(s, a^{\mathcal{C}_k}) + \epsilon_Q(s,a)$$

$$\leq V_\theta(s) + \frac{(\sum_{k=1}^{K} 2^{|\mathcal{C}_k|+1} - 2K)N(1-\gamma)\lambda}{d_\mu^{\pi_\theta}(s)|\mathcal{S}|} + 2K\left(|\epsilon_r| + \epsilon_P \gamma |S|/(1-\gamma)\right) \qquad \text{(by Lemma 6).}$$

Letting $\theta^*$ be the parameters of the optimal joint policy, we have

$$V_{\theta^*}(\mu) - V_\theta(\mu)$$

$$= \frac{1}{1-\gamma} \mathbb{E}_{\bar{s} \sim d_\mu^{\pi_{\theta^*}}} \mathbb{E}_{\bar{a} \sim \pi_{\theta^*}} \left[ A_\theta(\bar{s}, \bar{a}) \right] \qquad \text{(by performance difference lemma)}$$

$$\leq \frac{1}{1-\gamma} \mathbb{E}_{\bar{s} \sim d_\mu^{\pi_{\theta^*}}} \mathbb{E}_{\bar{a} \sim \pi_{\theta^*}} \left[ \frac{(\sum_{k=1}^{K} 2^{|\mathcal{C}_k|+1} - 2K)(1-\gamma)\lambda}{d_\mu^{\pi_\theta}(s)|\mathcal{S}|} + 2K\left(|\epsilon_r| + |\epsilon_P|\gamma|S|/(1-\gamma)\right) \right]$$

$$= \frac{1}{1-\gamma} \sum_{\bar{s}} d_\mu^{\pi_{\theta^*}}(\bar{s}) \left[ \frac{(\sum_{k=1}^{K} 2^{|\mathcal{C}_k|+1} - 2K)(1-\gamma)\lambda}{d_\mu^{\pi_\theta}(s)|\mathcal{S}|} \right] + 2K\left(|\epsilon_r|/(1-\gamma) + |\epsilon_P|\gamma|S|/(1-\gamma)^2\right)$$

$$\leq \frac{1}{1-\gamma} \sum_{\bar{s}} M \left[ \frac{(\sum_{k=1}^{K} 2^{|\mathcal{C}_k|+1} - 2K)(1-\gamma)\lambda}{|\mathcal{S}|} \right] + 2K\left(|\epsilon_r|/(1-\gamma) + |\epsilon_P|\gamma|S|/(1-\gamma)^2\right)$$

$$= \left( \sum_{k=1}^{K} 2^{|\mathcal{C}_k|+1} - 2K \right)\lambda M + 2K\left(|\epsilon_r|/(1-\gamma) + |\epsilon_P|\gamma|S|/(1-\gamma)^2\right).$$

Thus, we know that $(\pi_{\theta^1}^1, \cdots, \pi_{\theta^N}^N)$ is an $\left( (\sum_{k=1}^{K} 2^{|\mathcal{C}_k|+1} - 2K)\lambda M + 2K\left(|\epsilon_r|/(1-\gamma) + \gamma|S||\epsilon_P|/(1-\gamma)^2\right) \right)$-optimal policy. $\qquad \square$

## A.6 PROOF OF THEOREM 2

*Proof.* Since $L_\lambda(\theta)$ is $\beta_\lambda$-smooth, we have

$$\min_{t \leq T} \left\| \nabla_\theta L_\lambda(\theta^{(t)}) \right\|_2^2 \leq \frac{2\beta_\lambda(L_\lambda(\theta^*) - L_\lambda(\theta_0))}{T} \leq \frac{2\beta_\lambda(V_{\max} - V_{\min})}{T} \leq \frac{2\beta_\lambda}{T(1-\gamma)},$$

where the second inequality holds because we initialize $\theta_0 = 0$. We can choose $T$ large enough such that

$$\sqrt{\frac{2\beta_\lambda}{T(1-\gamma)}} \leq \lambda/(2|\mathcal{S}||\mathcal{A}| \max_i |\mathcal{A}^i|).$$

Solving the above inequality we obtain $T \geq \frac{8\beta_\lambda |\mathcal{S}|^2 |\mathcal{A}|^2 \max_i |\mathcal{A}^i|^2}{\lambda^2(1-\gamma)}$. By Lemma 3, we should set $\lambda = \frac{\epsilon}{(\sum_{k=1}^{K} 2^{|\mathcal{C}_k|+1} - 2K)M}$ to achieve the specified optimality-gap of $\epsilon + 2K\left(|\epsilon_r|/(1-\gamma) + \gamma|S||\epsilon_P|/(1-\right.$

$\gamma)^2$). Plugging in $\lambda = \frac{\epsilon}{(\sum_{k=1}^{K} 2^{|\mathcal{C}_k|+1} - 2K)M}$ and $\beta_\lambda := \frac{8N}{(1-\gamma)^3} + \frac{2\lambda N}{|\mathcal{S}|}$, we have

$$
\begin{aligned}
T \geq & \frac{8M^2 \beta_\lambda |\mathcal{S}|^2 |\mathcal{A}|^2 \max_i |\mathcal{A}^i|^2 (\sum_{k=1}^{K} 2^{|\mathcal{C}_k|+1} - 2K)^2}{\epsilon^2 (1-\gamma)} \\
= & \frac{64NM^2 |\mathcal{S}|^2 |\mathcal{A}|^2 \max_i |\mathcal{A}^i|^2 (\sum_{k=1}^{K} 2^{|\mathcal{C}_k|+1} - 2K)^2}{\epsilon^2 (1-\gamma)^4} + \frac{8M^2 \frac{2\lambda N}{|\mathcal{S}|} |\mathcal{S}|^2 |\mathcal{A}|^2 \max_i |\mathcal{A}^i|^2 (\sum_{k=1}^{K} 2^{|\mathcal{C}_k|+1} - 2K)^2}{\epsilon^2 (1-\gamma)} \\
= & \frac{256NM^2 |\mathcal{S}|^2 |\mathcal{A}|^2 \max_i |\mathcal{A}^i|^2 (\sum_{k=1}^{K} 2^{|\mathcal{C}_k|} - K)^2}{\epsilon^2 (1-\gamma)^4} + \frac{32NM |\mathcal{S}| |\mathcal{A}|^2 \max_i |\mathcal{A}^i|^2 (\sum_{k=1}^{K} 2^{|\mathcal{C}_k|} - K)}{\epsilon (1-\gamma)}
\end{aligned}
$$

$\square$

## A.7 PROOF OF PROPOSITION 1

*Proof.* Let $\{\mathcal{C}_k\}_{k=1}^{K}$ be the coarser partition and $\{\mathcal{C}'_{k'}\}_{k'=1}^{K'}$ be its refinement, i.e. $\mathcal{C}'_{k'} \subseteq \mathcal{C}_{\varphi(k')}$ for a mapping $\varphi : \{1, \ldots, K'\} \rightarrow \{1, \ldots, K\}$.

Because the Markov game is decomposed by $\{\mathcal{C}'_{k'}\}$ with errors $(\epsilon'_P, \epsilon'_r)$, by Definition 2 we have, for all $s, s' \in \mathcal{S}$ and $a \in \mathcal{A}$,

$$
P(s' \mid s, a) = \sum_{k'=1}^{K'} P^{k'}(s' \mid s, a^{\mathcal{C}'_{k'}}) + \epsilon'_P(s' \mid s, a), \qquad r(s, a) = \sum_{k'=1}^{K'} r^{k'}(s, a^{\mathcal{C}'_{k'}}) + \epsilon'_r(s, a).
\tag{13}
$$

**Constructing a decomposition for the coarser partition.** For each coarse block $\mathcal{C}_k$ define

$$
P^k(s' \mid s, a^{\mathcal{C}_k}) := \sum_{k' : \varphi(k')=k} P^{k'}(s' \mid s, a^{\mathcal{C}'_{k'}}), \qquad r^k(s, a^{\mathcal{C}_k}) := \sum_{k' : \varphi(k')=k} r^{k'}(s, a^{\mathcal{C}'_{k'}}).
$$

Summing over $k = 1, \ldots, K$ and substituting (13),

$$
\sum_{k=1}^{K} P^k(s' \mid s, a^{\mathcal{C}_k}) = \sum_{k'=1}^{K'} P^{k'}(s' \mid s, a^{\mathcal{C}'_{k'}}) = P(s' \mid s, a) - \epsilon'_P(s' \mid s, a),
$$

$$
\sum_{k=1}^{K} r^k(s, a^{\mathcal{C}_k}) = \sum_{k'=1}^{K'} r^{k'}(s, a^{\mathcal{C}'_{k'}}) = r(s, a) - \epsilon'_r(s, a).
$$

Hence $P$ and $r$ admit the coarse decomposition

$$
P(s' \mid s, a) = \sum_{k=1}^{K} P^k(s' \mid s, a^{\mathcal{C}_k}) + \underbrace{\epsilon'_P(s' \mid s, a)}_{=: \ \epsilon_P(s' \mid s, a)}, \quad r(s, a) = \sum_{k=1}^{K} r^k(s, a^{\mathcal{C}_k}) + \underbrace{\epsilon'_r(s, a)}_{=: \ \epsilon_r(s, a)}.
$$

**Error comparison.** Because we have simply *reused* the original error terms,

$$
|\epsilon_P| = \max_{s, s', a} |\epsilon_P(s' \mid s, a)| = \max_{s, s', a} |\epsilon'_P(s' \mid s, a)| = |\epsilon'_P|, \quad |\epsilon_r| = |\epsilon'_r|.
$$

Consequently $|\epsilon_P| \leq |\epsilon'_P|$ and $|\epsilon_r| \leq |\epsilon'_r|$, completing the proof. $\square$

## A.8 PROOF OF PROPOSITION 2

*Proof.* Recall $g(\{\mathcal{C}_k\}_{k=1}^{K}) := \sum_{k=1}^{K} 2^{|\mathcal{C}_k|} - K$. We show that splitting any block into two (thereby refining the partition) never increases $g$; applying this operation repeatedly proves monotonicity for an arbitrary refinement chain.

Let a partition $\{\mathcal{C}_k\}_{k=1}^{K}$ be given, and fix some $\mathcal{C}_1$ with $|\mathcal{C}_1| = m \geq 2$. Split it into two non-empty disjoint sets $\mathcal{A}, \mathcal{B}$ such that $\mathcal{A} \cup \mathcal{B} = \mathcal{C}_1$ and $|\mathcal{A}| = a$, $|\mathcal{B}| = b$ with $a, b \geq 1$ and $a + b = m$. The new (refined) partition therefore has $K + 1$ blocks, and its $g$-value is

$$
g_{\text{new}} = \left( 2^a + 2^b \right) + \sum_{k=2}^{K} 2^{|\mathcal{C}_k|} - (K+1).
$$

The change $\Delta := g_{\text{new}} - g_{\text{old}}$ satisfies

$$\Delta = \left(2^a + 2^b\right) - 2^m - 1 \quad \left(\text{since } g_{\text{old}} \text{ contains } 2^m - 1 \text{ for } \mathcal{C}_1\right)$$
$$= 2^a + 2^b - 2^{a+b} - 1.$$

Because $a, b \geq 1$, we have $2^a, 2^b \leq 2^{a+b-1}$, and therefore

$$2^a + 2^b \leq 2^{a+b-1} + 2^{a+b-1} = 2^{a+b}.$$

Therefore $\Delta \leq -1 \leq 0$. Equality is impossible, so $g$ *strictly decreases* after any non-trivial split.

Since any refinement can be obtained by a finite sequence of such splits, it follows that if $\{\mathcal{C}'_{k'}\}_{k'=1}^{K'}$ refines $\{\mathcal{C}_k\}_{k=1}^{K}$, then

$$g\left(\{\mathcal{C}'_{k'}\}\right) \leq g\left(\{\mathcal{C}_k\}\right).$$

$\square$

# B  EXPERIMENTS DETAILS

## B.1  ALGORITHM 1

---
**Algorithm 1** Dependency-based subteams construction under an edge budget

---
**Require:** Agents $\mathcal{N}$, dependency scores $d_{ij}$ for $i, j \in \mathcal{N}$, edge budget $B$
1: Initialize subsets $\{i\}$ for $i \in \mathcal{N}$ and edge set $\mathcal{E} \leftarrow \emptyset$        ▷ Singleton subteams with no edges
2: **while** there are more than one subset **and** $E < B$ **do**
3:      Find the two subsets $\mathcal{C}, \mathcal{C}'$ that maximize $d(\mathcal{C}, \mathcal{C}')$
4:      $\mathcal{E}_{\text{new}} \leftarrow \{(u, v) : u \in \mathcal{C}, v \in \mathcal{C}'\}$        ▷ All edges from $\mathcal{C}$ to $\mathcal{C}'$
5:      **if** $|\mathcal{E}| + |\mathcal{E}_{\text{new}}| > B$ **then** break        ▷ Budget would be exceeded
6:      $\mathcal{E} \leftarrow \mathcal{E} \cup \mathcal{E}_{\text{new}}$, merge $\mathcal{C}$ and $\mathcal{C}'$        ▷ Merge the two subteams
7: **end while**
8: **return** Subteams $\{\mathcal{C}_k\}_k$ partitioning a DAG $G = (\mathcal{N}, \mathcal{E})$

---

## B.2  DETAILS OF DECOMPOSITION ERROR FITTING

We measure the decomposition errors $|\epsilon_P|$ and $|\epsilon_r|$ (cf. Definition 2) by fitting local models $\{P^k, r^k\}$ for each subteam $\mathcal{C}_k$. The goal is to estimate how well the environment's dynamics and rewards can be factorized according to different subteam partitions.

### B.2.1  NEURAL NETWORK ARCHITECTURE

For each subteam $\mathcal{C}_k$, we implement two multi-layer perceptron models:

- A transition model $P^k_{\psi_k} : \mathcal{A}^{\mathcal{C}_k} \to \mathbb{R}^{|\mathcal{S}|}$ that maps subteam actions to transition components
- A reward model $r^k_{\phi_k} : \mathcal{S} \to \mathbb{R}$ that maps subteam states to reward components

Both networks use a three-layer architecture with hidden dimension 128:

$$\text{FC}(\text{in\_dim}, 128) \to \text{ReLU} \to \text{FC}(128, 128) \to \text{ReLU} \to \text{FC}(128, \text{out\_dim})$$

### B.2.2  GLOBAL APPROXIMATION

We reconstruct global approximations by summing the outputs of the subteam-specific networks:

$$\widehat{P}_\psi(\cdot|a) = \sum_k P^k_{\psi_k}(\cdot|a^{\mathcal{C}_k}), \quad \widehat{r}_\phi(s) = \sum_k r^k_{\phi_k}(s).$$

The decomposition errors are then computed as the maximum absolute differences between the true environment dynamics and our factorized approximations:

$$|\widehat{\epsilon_P}| = \max_{a, s'} |\widehat{P}_\psi(s'|a) - P(s'|a)|, \quad |\widehat{\epsilon_r}| = \max_s |\widehat{r}_\phi(s) - r(s)|.$$

### B.2.3  TRAINING PROCEDURE

The models for each partition structure were trained using the Adam optimizer with a learning rate of $10^{-3}$ over 10,000 epochs. The training process minimizes two Mean Squared Error losses:

$$\mathcal{L}_P(\psi) = \frac{1}{|\mathcal{A}||\mathcal{S}|} \sum_{a \in \mathcal{A}, s' \in \mathcal{S}} \left( P(s'|a) - \sum_k P^k_{\psi_k}(s'|a^{\mathcal{C}_k}) \right)^2 \tag{14}$$

$$\mathcal{L}_r(\phi) = \frac{1}{|\mathcal{S}|} \sum_{s \in \mathcal{S}} \left( r(s) - \sum_k r^k_{\phi_k}(s^{\mathcal{C}_k}) \right)^2 \tag{15}$$

## B.3 DYNAMIC DAG CONSTRUCTION

**Handling variable parents.** To accommodate the variable parent sets $\mathcal{P}^i$ induced by the dynamic DAG at each timestep, we construct the input to agent $i$'s actor as the concatenation of its own observation (or encoded state) and the actions (and optionally observations) of its parents:

$$\text{Input}_i = \text{Concat}(o^i \text{ or } \phi(o^i), \ \{a^j, o^j\}_{j \in \mathcal{P}^i}),$$

where $\phi(\cdot)$ is an optional encoder. For agents that are not parents of agent $i$, their actions are zero-padded to ensure a fixed input dimension. This design enables consistent batching across agents and supports seamless integration into standard actor-critic architectures.

**Dependency score computation.** The pairwise dependency scores $d_{ij}$ reflect the spatial proximity between agents and are computed as the negative pairwise distances, with specific formulations for each environment:

- **Coordination Game:**
$$d_{ij} = -|s^i - s^j|,$$
where $s^i \in \{0, 1\}$ denotes the binary state of agent $i$.

- **Aloha:** Agents are fixed on a $2 \times 5$ grid at positions $(x_i, y_i) \in \{0, 1\} \times \{0, 1, 2, 3, 4\}$. The dependency score is the negative Manhattan distance:
$$d_{ij} = -(|x_i - x_j| + |y_i - y_j|).$$

- **Predator-Prey, Warehouse, and Battle:** Each agent $i$ occupies a continuous 2D position $(x_i, y_i) \in \mathbb{R}^2$. The dependency score is the negative Euclidean distance:
$$d_{ij} = -\sqrt{(x_i - x_j)^2 + (y_i - y_j)^2}.$$

## B.4 PSEUDOCODE FOR THE REWARD FUNCTION IN COORDINATION GAME

---
**Algorithm 2** Compute Team Reward for $N$ Agents in State $s$

---
1: **if** $N \in \{2, 3\}$ **then**
2:     `difference_bound` $\leftarrow 1$
3: **else**
4:     `difference_bound` $\leftarrow 2$
5: **end if**
6: $c_0 \leftarrow s.\text{count}(0)$
7: $c_1 \leftarrow s.\text{count}(1)$
8: **if** $|c_0 - c_1| \leq$ `difference_bound` **then**
9:     **if** $c_0 < c_1$ **then**
10:         `reward` $\leftarrow 1$
11:     **else**
12:         `reward` $\leftarrow 0$
13:     **end if**
14: **else if** $c_0 > c_1$ **then**
15:     `reward` $\leftarrow 3$
16: **else**
17:     `reward` $\leftarrow 2$
18: **end if**

---

## B.5 HYPERPARAMETERS

Table 2: Hyperparameters for MAPPO (Coordination Game and Aloha), MADDPG (Predator-Prey)

| Hyperparameter | Value |
| --- | --- |
| Environment steps | $2 \times 10^5$ (CG), $5 \times 10^5$ (Aloha), $4 \times 10^5$ (Predator-Prey) |
| Episode length | 20 (CG), 25 (Aloha and Predator-Prey) |
| PPO epochs | 5 (CG and Aloha) |
| Actor/Critic learning rate | $7 \times 10^{-4}$ (CG and Aloha), $1 \times 10^{-2}$ (Predator-Prey) |
| Optimizer | Adam |
| Evaluation episodes | 100 (CG and Aloha), 200 (Predator-Prey) |
| Rollout threads | 32 (CG and Aloha) |
| Training threads | 32 (CG and Aloha) |
| Hidden size | 64 (CG and Aloha), 128 (Predator-Prey) |
| Random seeds | 60 (CG and Aloha), 10 (Predator-Prey) |
| Actor architecture (CG) | $\mathrm{Concat}(\texttt{Base}(s), a^{\mathcal{P}^i}) \to \mathrm{FC}(|\mathcal{A}^i|) \to \mathrm{Softmax}$ |
| Actor architecture (Aloha) | $\mathrm{Concat}(\texttt{Base}(o^i), \texttt{Base}(\mathrm{Concat}(o^{\mathcal{P}^i}, a^{\mathcal{P}^i}))) \to \mathrm{FC}(|\mathcal{A}^i|) \to \mathrm{Softmax}$ |
| Actor architecture (Predator-Prey) | $\mathrm{Concat}(o^i, a^{\mathcal{P}^i}) \to \mathrm{FC} \to \mathrm{ReLU} \to \mathrm{FC} \to \mathrm{ReLU} \to \mathrm{FC}(|\mathcal{A}^i|)$ |
| Critic architecture (CG/Aloha) | Joint state or observation $\to \texttt{Base} \to \mathrm{FC}(1)$ |
| Critic architecture (Predator-Prey) | $[o^i; a^i]_{i \in \mathcal{N}} \to \mathrm{GCN}_1 \to \mathrm{FC} \to \mathrm{GCN}_2 \to \mathrm{FC} \to \mathrm{MaxPool} \to \mathrm{FC}(1)$ |

**Notation:** CG = Coordination Game. `Base`: $\mathrm{FC(hidden)} \to \mathrm{ReLU} \to \mathrm{FC(hidden)} \to \mathrm{ReLU}$

Table 3: Hyperparameters for value-based methods (Warehouse and Battle)

| Hyperparameter | Value |
| --- | --- |
| Environment steps | $1.5 \times 10^6$ (Warehouse), $2 \times 10^6$ (Battle) |
| Episode length | 50 (Warehouse), 100 (Battle) |
| Learning rate | $1 \times 10^{-3}$ |
| Optimizer | Adam |
| Evaluation episodes | 10 |
| Hidden size | 64 (Subteam $Q^{\mathcal{C}_k}$), 128 (Mixer) |
| Random seeds | 10 (Warehouse), 5 (Battle) |
| Subteam ratio $\eta$ | $1/4$ ($K = \lceil \eta N \rceil$) |
| Subteam $Q^{\mathcal{C}_k}$ architecture (per subteam $\mathcal{C}_k$) | $s \to \mathrm{FC} \to \mathrm{ELU} \to \mathrm{FC}(64) \to \mathrm{ELU} \to \mathrm{FC}(|\mathcal{A}^{\mathcal{C}_k}|)$ |
| Mixer architecture (QTRAN/VAST) | $\mathrm{Concat}(Q^{\mathcal{C}_1}, \ldots, Q^{\mathcal{C}_K}) \to \mathrm{FC}(128) \to \mathrm{ELU} \to \mathrm{FC}(128) \to \mathrm{ELU} \to \mathrm{FC}(1)$ |

**Notes:** All hyperparameters follow VAST (Phan et al., 2021b).

## C  ADDITIONAL RESULTS

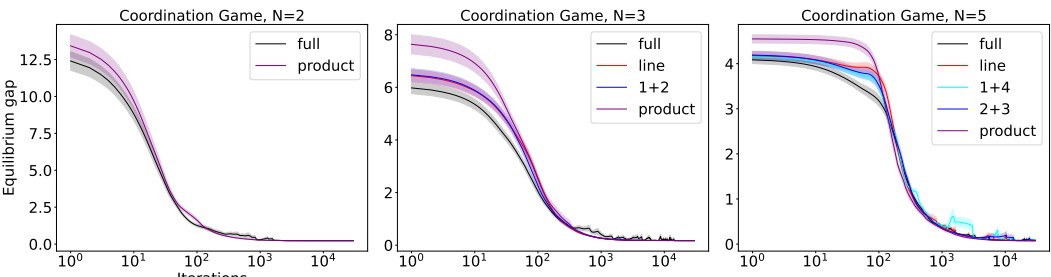

Figure 4: Equilibrium gap of tabular softmax BN policy gradient ascent under various DAG topologies. Averaged over 50 seeds, with shaded areas showing standard error; initial policy $\theta_0 \sim \mathcal{N}(0, 1)$.

# D    RESULTS ON SMAC

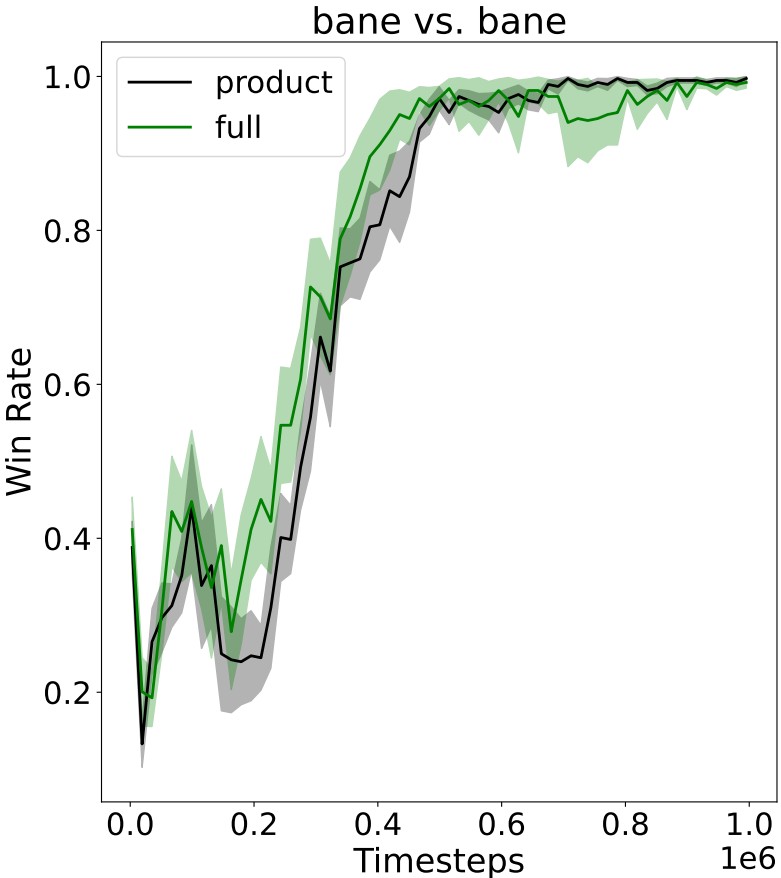

Figure 5: Full DAG vs. product DAG with MAPPO on the SMAC bane vs. bane map.

The learning curves in Figure 5 display the mean and standard error over 12 random seeds on the SMAC bane vs. bane map, a heterogeneous large-scale scenario with 24 controllable agents (20 Zerglings and 4 Banelings). The fully correlated DAG and the product DAG achieve nearly identical performance, indicating that additional correlation provides little benefit in this environment and therefore the heuristic is not expected to outperform the product structure when even the fully expressive DAG offers no advantage.

