# OpenReview forum: "Correlated Policy Optimization in Multi-Agent Subteams"
_ICLR.cc/2026/Conference — ICLR 2026 Poster_

### Official Review · Reviewer_2vBZ · 2025-10-30

**Soundness:** 3
**Presentation:** 3
**Contribution:** 2
**Rating:** 6
**Confidence:** 3

**Summary:**

The paper considers subteams in cooperative multi-agent RL as a means to balance coordination and scalability. First, under the tabular setting, the authors establish theoretical results investigating convergence and suboptimality for bayesian networks policy gradients under decomposability assumptions. Second, the authors empirically validate their theoretical results in a tabular setting. Third, the authors propose a practical approach for constructing subteams in MARL tasks and test it in three MARL environments, as well as investigate how it can be used for value factorization in centralized training.

**Strengths:**

The papers features a good discussion of related work in Sec. 2. I believe the key contributions of this work are related with the analysis of gradient-based methods to optimize BN policies - the paper seems to contribute with a novel analysis of gradient-based methods to optimize BN policies in the tabular setting. The paper is clear and well-organized. The experiments seem to support the theoretical findings.

**Weaknesses:**

The contribution of the paper in the empirical part seems a bit marginal. In particular, I have two key concerns (take a look at my last question below for additional details): (i) are the baselines considered representative of the multiple previously proposed algorithms discussed in Sec. 2?; and (ii) the empirical algorithms analysed in Sec. 6.2 do not seem directly linked to the algorithm analyzed in the theoretical part of the paper. The theoretical results also rely on some assumptions that are a bit restrictive (access to the underlying state and Asssumption 1), but I believe this should not be a reason to reject the paper alone.

**Questions:**

- The authors consider access to the full underlying state of the environment (Markov game). What about the case of Dec-POMDPs?
- While assumption 1 may be used by previous works, I still believe it is a bit restrictive. How hard would it be to extend the theoretical results while removing Assumption 1?
- Maybe the authors could discuss after presenting Theo. 1 what are the key differences between the result in Theo. 1 and the equivalent single-agent result.
- line 288 - "1. Therefore, the suboptimality bound in (6) reveals a tradeoff when choosing the fineness/coarseness of the decomposition" - Is it possible to find the optimal tradeoff between the two terms in the bound of Lemma 3?
- How many runs were performed to get the empirical results (e.g., Fig. 1)?
- Figure 1: What do the shaded areas in the plots represent?
- Secs. 6.2 and 6.3 of the paper seem a bit disconnected from the theoretical part of the paper. I'm particularly referring to the algorithmic aspect (not to the problem of finding good partitions). The gradient ascent algorithm proposed and analysed by the authors in (2) and (3) considers a regularized objective. What would be the closest "function-approximation" variant of such an algorithm? Why did you choose MAPPO and MADDPG? Can these algorithms be seen as some kind of "function-approximation-counterparts" of the tabular algorithm in (2)-(3)?
- No other previous works (e.g., those discussed in Sec. 2) propose ways to perform subteams decomposition? If so, I believe such methods should also be included as baselines in the empirical part of the work.

---

> ### Author Response · Authors · 2025-11-26
> **Rebuttal (1/2)**
>
> Thank you for carefully reviewing our paper!
> Please find below our response to the Weaknesses and Questions.
>
> **W1/Q8: The choice of the baselines**
>
> Until Sec. 6.2.2, our paper has exclusively focused on optimizing Bayesian network (BN) policies.To our knowledge, no prior works have studied subteams decomposition for BN policy optimization.
> Prior works as discussed in Sec. 2 mostly operate in the CTDE framework, as they perform subteams decomposition when learning centralized critics while optimizing product policies. Those methods cannot be readily repurposed for optimizing BN policies.
> We noticed our heuristic method could be repurposed for subteams decomposition in CTDE and therefore conducted Sec. 6.2.2 to show the generalizability of our theoretical insight, which complemented the main claims of the paper (on optimizing BN policies).
>
> **W2/Q7: Secs. 6.2 vs the theoretical part, the "function-approximation" variant**
>
> As Section 6.1 strictly adheres to the theoretical part, the closest "function-approximation" variant would be to replace the tabular softmax policy parameterization in Section 6.1 with a function approximator like neural networks (i.e., “actor networks”), where the regularizer can be straightforwardly applied in a similar fashion as adding an “entropy loss” term.
> Although legitimate, we felt such a closest variant might be too incremental from Section 6.1, and readers might be eager to see a more practical setting like we did in Secs. 6.2.1 with MAPPO and MADDPG, which also use actor networks and entropy regularization and therefore we would view them as valid "function-approximation-counterparts" of the tabular algorithm in (2)-(3).
>
> **Q1: Full/partial observability, extension to Dec-POMDPs**
>
> Full observability has been a standard assumption in prior theoretical works, including the closely related ones regarding product policy [1,2,3] and BN policy [4].
> Relaxing it introduces major challenges to these works. For example,
> - The notion of equilibrium and suboptimality would need to be re-defined non-trivially under partial observability.
> - It is known that solving a Dec-POMDP is (PSPACE-)hard, because the number of histories/beliefs can grow exponentially with the number of states and the (effective) horizon, while the optimal policy is in general history-dependent.
>
> Under partial observability, some prior works only provide asymptotic convergence guarantees by treating histories as states (e.g., [5]). However, by b), this histories-as-states method only yields exponential convergence rates, unless additional assumptions on the problem structure are given.
> For example, the “rich-observation” structure is assumed in [6]; a recent work [7] only considers Dec-POMDPs with a factored structure and memoryless policy.
> This suggests that fully addressing partial observability for our motivation easily requires another full paper.
>
> **Q2: Hardness of relaxing Assumption 1**
>
> Assumption 1 guarantees that the state-visitation mismatch is finite and bounded by $M$, which is required in Lemma 2 and Lemma 3. Relaxing Assumption 1 would require a different method for controlling the visitation-distribution shift between policies, or one needs an entirely different mechanism to establish the relationship between stationary points and Nash Equilibria, which is technically nontrivial.
>
> **Q3: Difference between Theorem 1 and the single-agent counterpart**
>
> In the single-agent setting [8], the established result guarantees convergence to the optimal softmax policy under the log-barrier regularizer. In our setting, we cannot always treat all agents as a single decision maker, unless they are fully correlated. Consequently, Theorem 1 can only guarantee convergence to an equilibrium, which in general need not be globally optimal. When the BN is fully correlated ($K=1$), Theorem 2 shows that the convergent policy, which is an equilibrium policy by Theorem 1, is in fact optimal, matching the single-agent result.
>
> We also note that the asymptotic result in [4] for the unregularized policy gradient guarantees only convergence to an equilibrium, even under fully correlated BNs. The reason is that, without regularization, parent actions are not sufficiently explored, so optimality cannot be ensured. In contrast, the log-barrier regularizer guarantees adequate exploration, which allows our fully correlated case ($K=1$) to recover optimal convergence.

---

> ### Author Response · Authors · 2025-11-26
> **Rebuttal (2/2)**
>
> **Q4: Optimal bias–convergence tradeoff in bound (line 288)**
>
> Yes, there would be a theoretically optimal partition that minimizes the sum of the two terms in (6).
> However, the decomposition errors $|\epsilon_r|$ and $|\epsilon_P|$ are not available in closed form (and are generally hard to compute exactly), so an optimal partition is not expressible analytically.
>
> **Q5: The number of runs; shaded areas**
>
> This information was provided in Table 2 and 3 in the appendix of the original submission. We have
> - 50 seeds for each curve in Figure 1.
> - 60, 60, and 10 seeds for Coordination Game, Aloha, and Predator-Prey, respectively, in Figure 2.
> - 10 and 5 seeds for Warehouse and Battle, respectively, in Figure 3.
> - The shaded areas represent the standard errors of the means.
> Our revision has modified the captions of Figures 1,2, and 3 to include the number of runs and what the shaded areas represent.
>
>
> [1] Leonardos et al., "Global Convergence of Multi-Agent Policy Gradient in Markov Potential Games", ICLR 2022
>
> [2] Zhang et al., "Gradient play in stochastic games: stationary points, convergence, and sample complexity", IEEE Transactions on Automatic Control, 2024
>
> [3] Zhang et al., "On the Global Convergence Rates of Decentralized Softmax Gradient Play in Markov Potential Games", NeurIPS 2022
>
> [4] Chen et al., "Context-Aware Bayesian Network Actor-Critic Methods for Cooperative Multi-Agent Reinforcement Learning", ICML 2023
>
> [5] Zhang et al., "FOP: Factorizing Optimal Joint Policy of Maximum-Entropy Multi-Agent Reinforcement Learning", ICML 2021
>
> [6] Zang et al., "Automatic Grouping for Efficient Cooperative Multi-Agent Reinforcement Learning", NeurIPS 2023
>
> [7] Gai, Jingchu, Qinghua Liu, Zhuoran Yang, and Chi Jin. "Provable Learning for DEC-POMDPs: Factored Models and Memoryless Agents", ICLR 2025 submission.
>
> [8] Agarwal et al., "On the theory of policy gradient methods", JMLR 2021

---

### Official Review · Reviewer_g5cJ · 2025-10-31

**Soundness:** 4
**Presentation:** 4
**Contribution:** 3
**Rating:** 8
**Confidence:** 3

**Summary:**

This paper studies correlated policy optimization in cooperative multi-agent reinforcement learning (MARL) using subteam-based coordination. The authors represent joint policies as Bayesian networks (BNs), where agents within a subteam are fully connected and thus act in a correlated manner, while inter-subteam dependencies are limited.

Theoretical results extend previous work on BN policy gradients by establishing finite-time convergence to near-optimal policies under a decomposability assumption on the environment’s reward and transition functions. The analysis reveals a trade-off: finer partitions of agents yield faster convergence but potentially higher suboptimality due to decomposition errors.

Empirically, the authors validate their theory in a tabular coordination game that precisely satisfies the theoretical assumptions, and then introduce a heuristic for constructing context-aware subteams based on pairwise dependency scores. This heuristic is integrated into deep MARL algorithms such as MAPPO and MADDPG, where it consistently improves learning speed and final performance across multiple benchmark environments.

**Strengths:**

- The problem is well-motivated. The work addresses a central challenge in MARL—scaling coordination to large agent populations—by introducing a principled way to exploit structured correlations.
- The exposition is clear, and the theoretical intuitions are well communicated.
- The paper has a strong theoretical contribution.  The finite-time convergence and near-optimality guarantees under decomposability assumptions extend prior results (e.g., Chen & Zhang, 2023) in a meaningful way.
- The experiments are well-aligned with the theory. The tabular experiments are well designed to exactly satisfy the theoretical setup, lending credibility to the claims.
- The empirical results are convincing. The proposed heuristic for subteam construction yields consistent performance improvements in deep MARL tasks, as shown in Figures 2–3.

**Weaknesses:**

Dependence on domain-specific priors: The heuristic requires a priori pairwise dependency scores dij , which rely on domain knowledge. This may limit generality and make comparisons to methods that learn dependency structures less direct.

**Questions:**

1. How exactly are the dij dependency scores computed in each environment? Are they purely distance-based, or do they involve any learned or empirical statistics?
2. In Section 6.2.1 you mention that “parent actions are detached from the computation graph to prevent backpropagation.” Could you elaborate on this?
3. Have the authors seen or considered the following closely related paper? Kapoor, A., Freed, B., Schneider, J., & Choset, H. (2025). Assigning Credit with Partial Reward Decoupling in Multi-Agent Proximal Policy Optimization. Reinforcement Learning Journal, 1, 380–399. This work also decomposes agents into subgroups with partially decoupled rewards and might provide a useful empirical or conceptual comparison.

---

> ### Author Response · Authors · 2025-11-26
>
> Thank you for carefully reviewing our paper!
> Please find below our response to the Weaknesses and Questions.
>
> **Q1: The dependency scores**
>
> We provided the detailed computation of the dependency scores in Appendix B.3. Specifically, in the Coordination Game, the score is not distance-based; instead, it is derived directly from the agents’ local binary states. In Aloha, the agents are arranged in a 2d grid and we use Manhattan distance. In Predator–Prey, Warehouse, and Battle, we use Euclidean distance based on agents’ spatial coordinates.
>
> We agree this dependency on domain priors may limit generality.
> In our experiments, we intentionally used generic, low-engineering dependency proxies rather than task-tuned scores. Despite this, the resulting subteam heuristic yields consistent gains and compares favorably with learning-based structure methods (e.g., VAST variants) This suggests that even weak priors can be effective when aligned with the subteam-decomposability motivation.
>
>
> **Q2: Detaching parent actions**
>
> When updating an agent’s policy, we treat the actions of its parents as fixed inputs rather than differentiable quantities (in PyTorch, by applying $\texttt{detach()}$ to the parent actions), so the loss defined for this agent does not get propagated to its parents through the parent actions.
>
> **Q3: Comparison with Kapoor et al. (2025)**
>
> Thank you for pointing out Kapoor et al. (2025). We were not aware of this work and have added a citation in Section 2 in the revision. Conceptually, it is related in that it decomposes agents into subgroups to improve multi-agent PPO credit assignment via partially decoupled rewards/advantages. Our approach is complementary: we focus on structured policy correlation via BN-induced DAGs and provide finite-time convergence and near-optimality guarantees under decomposability.
>
> We will add discussion in the related-work section clarifying both the connection and differences. An empirical comparison is potentially informative. However, Kapoor et al. (2025) uses soft attention without an explicit edge budget, so a strictly budget-matched comparison would require adapting their method to the same DAG/edge constraints.

---

### Official Review · Reviewer_Duum · 2025-10-31

**Soundness:** 2
**Presentation:** 3
**Contribution:** 2
**Rating:** 6
**Confidence:** 4

**Summary:**

This paper utilizes Bayesian Networks (BNs) to model the structured correlations among multi-agent subteams and proposes a class of correlated joint policies induced by directed acyclic graphs (DAGs). Building upon prior work, it establishes a convergence rate for tabular softmax BN policy gradient ascent under fixed DAG structures. Furthermore, the authors prove that, for BNs aligned with the context of multi-agent subteams, regularized policy gradient ascent converges to a policy with bounded suboptimality. Finally, this paper relaxes certain assumptions and proposes a heuristic method, which is evaluated across multiple benchmark environments.

**Strengths:**

1. The paper is well-structured, with a clear and well-defined motivation.

2. The description of the background and the theoretical analyses are detailed, with clear explanation of the underlying thoughts.

3. This is the first work that establishes optimality guarantees for BN policies without requiring full independence among agents.

4. This paper conducts sufficient experiments to demonstrate the effectiveness of the method.

**Weaknesses:**

1. The assumptions required for the proof section (full observability, fully correlated with a sub-team and fully independent across sub-teams, fixed DAG topology) are too strict.

2. Practical methods relax some assumptions in the proof section, but some may deviate significantly from the original proof (such as global observability).

3. Experimental scenarios are too simple to demonstrate the upper limit of the method's advantages.

**Questions:**

1. If possible, please answer the questions mentioned in the Weaknesses Section first.

2. What impact will partial observability have on the theoretical conclusions of this paper?

3. Why is maximizing the average pair-wise dependency score between the agents considered a reasonable criterion for merging two subteams? What are the underlying motivations or theoretical considerations behind this method?

4. What are the considerations for selecting baseline algorithms (MAPPO/MADDPG) in different experimental environments?

5. This is my second time reviewing your submission. As you can see, I continue to hold a generally positive view of your work. I appreciate that you have added new experimental scenarios in response to the previous round of reviewers’ comments — that is definitely a constructive improvement. However, it is somewhat disappointing that the additional experiments are still based on the VAST (2021) scenario, which is relatively dated and not particularly convincing as an evaluation benchmark. Therefore, the new results do not substantially strengthen the empirical evidence of your claims. I strongly encourage you to follow the suggestion of Reviewer QYoJ from NeurIPS25 and include experiments on large-scale scenarios in SMAC with more than 20 agents, which would make your framework’s advantages much more persuasive. For this round, I maintain my overall positive inclination toward the paper, but I must note that I cannot guarantee other reviewers will share the same perspective — some may weigh empirical validation more heavily than theoretical contribution. I therefore reserve the right to align my final assessment with the consensus on other reviewers' evaluation on your experiment.

---

> ### Author Response · Authors · 2025-11-26
> **Rebuttal (1/2)**
>
> Hi, We sincerely thank you for reviewing our work again and still being positive.
>
> We first address your new comment, i.e., Question 5, and then the old comments, i.e., Weaknesses 1-3 and Questions 1-4.
> Our responses to the old comments are largely repetitive, as they seemed to have addressed most of the concerns in the previous round of reviewing.
>
> **(Q5) The new comment**
> Following your and Reviewer QYoJ's suggestion,  we evaluate our framework on both the VAST scenarios and the SMAC scenarios.For the VAST scenarios, we observed a performance improvement by introducing agents subgroups/correlations, which led to Section 6.2.2.
>
> For the SMAC scenarios with more than 20 agents, we observed limited difference between $\texttt{product}$ and $\texttt{full}$, as shown in Figure 5 in Appendix D of our revision. A representative example is for the scenario of bane_vs_bane map, a heterogeneous large-scale scenario with 24 controllable agents (20 Zerglings and 4 Banelings).
> This suggests that for such SMAC scenarios, although widely considered as more complex, they have weak action-dependency structure, and thus any sub-budget DAG (including ours) cannot yield large gains. This empirical finding is informative and consistent with Theorem 2: when decomposition/cross-team dependence is near-zero, factorized policies are already near-optimal.
>
> Conversely, in benchmarks where meaningful dependencies exist and the induced decomposition errors are non-negligible, $\texttt{full}$ substantially outperforms $\texttt{product}$. Our heuristic reliably closes that gap under a limited edge budget. These are precisely the settings our framework is designed for, and they are validated across the other benchmarks in Sec. 6.
>
> **(W1, Q2) Full/partial observability**
> We agree that partial observability changes the theoretical landscape. Our near-optimality guarantee (Theorem 2) relies on Markov state access to define (i) the decomposability errors and (ii) the value-function decomposition underpinning the analysis.
>
> Full observability has been a common assumption in prior theoretical works, including the closely related ones regarding product policy [1,2,3] and BN policy [4].
> Relaxing it introduces major challenges to these works. For example,
> - The notion of equilibrium and suboptimality would need to be re-defined non-trivially under partial observability.
> - It is known that solving a Dec-POMDP is (PSPACE-)hard, because the number of histories/beliefs can grow exponentially with the number of states and the (effective) horizon, while the optimal policy is in general history-dependent.
>
> Existing Dec-POMDP theory either provides asymptotic guarantees by treating histories as states[5] or imposes strong structural conditions (e.g., rich observations [6], factored models [7]), which are orthogonal to our current focus.
>
> The core intuition of our approach is that reducing cross-subteam dependence can control approximation bias. We believe this intuition remains relevant when partial observability still yields approximately factorable interaction structure over beliefs/histories.
>
> **(W1) The other assumptions**
> While the other assumptions are seemingly restrictive, they are more relaxed than prior works. For example, our subteam assumptions subsumes those required to establish suboptimality guarantees for product policy and BN policy.
> Furthermore, our assumptions are indeed critical in the proofs: e.g., a key step in Lemma 3’s proof is to upper bound the gain of $a^{\mathcal C_k}$ over the subteam baseline $V_\theta^{k}(s)$, which derived from a telescoping sum that would fail if subteams are not independent (see Line 1050 in Appendix A.5 for details).
>
> **(W2) Our practical method vs proof**
> Experiments in Section 6.1 (tabular) adheres to all the theoretical assumptions, providing empirical evaluation of the theorems.
> Section 6.2 (deep RL) relaxes (and therefore deviates from) certain assumptions for practicality. Specifically, we (i) relax the requirement of global observability, and (ii) update the DAG intermittently using Algorithm 1 rather than fixing it. While these choices move beyond the formal proof setting, they preserve the main theoretical motivation: under a limited edge budget, search for partitions/DAGs that minimize cross-subteam dependence (i.e., decomposition error).

---

> ### Author Response · Authors · 2025-11-26
> **Rebuttal (2/2)**
>
> **(W3) On the experiments**
> We provide the following clarifications for interpreting our empirical study:
> The chosen benchmarks and base algorithms (MAPPO/MADDPG) are standard in the multi-agent coordination literature.
> Commensurate with the theorems, our experiments focus only on evaluating various strategies for policy correlation via subteams decomposition. For those benchmarks and base algorithms, the improvement by introducing policy correlation is upper bounded by the $\texttt{full}$-$\texttt{product}$ gap. Whether this gap is small or large, it serves as a reference when interpreting performance improvement.
> Theoretically, more DAG edges introduce more policy correlation and better performance. So, in the experiments we compare curves under a certain edge budget.
> Our method consistently outperforms $\texttt{product}$ and $\texttt{random}$ of the same budget across all environments, and matches or exceeds the full baseline in most cases, despite using fewer edges.
>
> **Q3 - On average pair-wise dependency**
> The dependency score quantifies the degree of interaction or influence between agents.
> Merging agents with high mutual dependencies into the same subteam aligns with the theoretical motivation of our work, which assumes stronger intra-subteam coordination and weaker inter-subteam interaction.
>
> Our specific criterion is “pair-wise” because the dependency score is assumed to be given per agent pair (e.g., physical distance in Predator-prey). It is “averaged” because merging larger subteams needs more edges and we want to efficiently use the edge budget $B$. The result is a computationally efficient heuristic.
>
> **Q4 - On MAPPO/MADDPG**
> MAPPO and MADDPG are widely regarded as state-of-the-art algorithms for these benchmarks.
> We select MAPPO for Coordination Game and Aloha as they involve discrete action spaces, aligning with [4]. Predator-Prey involves continuous action spaces, so we adopt MADDPG suited for that, following [5,6].
>
>
> [1] Leonardos et al., "Global Convergence of Multi-Agent Policy Gradient in Markov Potential Games", ICLR 2022
>
> [2] Zhang et al., "Gradient play in stochastic games: stationary points, convergence, and sample complexity", IEEE Transactions on Automatic Control, 2024
>
> [3] Zhang et al., "On the Global Convergence Rates of Decentralized Softmax Gradient Play in Markov Potential Games", NeurIPS 2022
>
> [4] Chen et al., "Context-Aware Bayesian Network Actor-Critic Methods for Cooperative Multi-Agent Reinforcement Learning", ICML 2023
>
> [5] Zhang et al., "FOP: Factorizing Optimal Joint Policy of Maximum-Entropy Multi-Agent Reinforcement Learning", ICML 2021
>
> [6] Zang et al., "Automatic Grouping for Efficient Cooperative Multi-Agent Reinforcement Learning", NeurIPS 2023
>
>
> [7] Gai, Jingchu, Qinghua Liu, Zhuoran Yang, and Chi Jin. "Provable Learning for DEC-POMDPs: Factored Models and Memoryless Agents", ICLR 2025 submission.

---

### Official Review · Reviewer_Qy1s · 2025-10-31

**Soundness:** 4
**Presentation:** 4
**Contribution:** 4
**Rating:** 8
**Confidence:** 3

**Summary:**

This paper addresses the critical challenge of scalability in cooperative multi-agent reinforcement learning (MARL), which is often hindered by the exponential growth of the joint action space. The authors draw inspiration from human teams to propose a "subteam-based coordination" structure. This structure is formalized using Bayesian networks (BNs) , where agents within a subteam are fully correlated (modeled as a fully connected subgraph ) while interactions between subteams are limited (modeled as conditionally independent ).

The paper's contributions are threefold:

- It provides a finite-time convergence rate for tabular softmax BN policy gradient ascent under a fixed DAG, strengthening prior asymptotic results.

- It proves that for this subteam-based BN structure, regularized policy gradient ascent converges to a near-optimal policy (not just an equilibrium) under a "decomposability condition" on the environment's reward and transition functions. The resulting suboptimality bound is explicitly characterized by the decomposition errors.


- It introduces a practical heuristic for dynamically constructing these subteam DAGs based on "dependency scores" and an edge budget. This heuristic is integrated with deep MARL algorithms (MAPPO/MADDPG) and value-factorization methods (VAST) , demonstrating superior performance over baselines across several benchmarks.

**Strengths:**

**Clear Motivation and Formalism**: The paper's premise is intuitive and well-motivated. The core idea of partitioning agents into highly-coordinated subteams to reduce complexity is a natural one. The use of Bayesian networks to formalize this—capturing full correlation within teams and independence between them—is an elegant and clean way to model the problem .

**Theoretical Contribution**: The main theoretical result (Theorem 2) is a strong contribution. It moves beyond typical convergence guarantees (to an equilibrium, like in Theorem 1) to provide a bound on the policy's suboptimality. Crucially, it connects this bound directly to a structural property of the environment: the decomposability errors. This provides a concrete, formal basis for why and when subteam factorization should work.

- **Identification of a Key Trade-off**: The theory and experiments jointly expose a fundamental trade-off in subteam design. Coarser partitions (i.e., fewer, larger subteams) lead to smaller decomposition errors and thus a lower asymptotic suboptimality bias. Finer partitions (i.e., more, smaller subteams) may converge faster but suffer from larger errors. The tabular experiments in Section 6.1, which fit the decomposition errors (Table 1) and plot the final suboptimality (Figure 1), provide a clear and compelling validation of this theoretical insight.

- **Empirical Validation and Practical Heuristic**: The paper successfully bridges the theoretical insights to a practical setting. The theory motivates the search for partitions with low decomposition error, and the authors propose a sensible heuristic (Algorithm 1) to approximate this by merging subteams with high "dependency" . The empirical results are comprehensive, showing the heuristic's effectiveness in:

  - Tabular settings (validating the theory).

  - Deep MARL actor-critic methods (outperforming 'full', 'product', and 'random' DAGs) .

  - Centralized training (CTDE) value-factorization methods (improving VAST) .

**Weaknesses:**

- **On the "Decomposability" Assumption**: The main result hinges on the environment's decomposability (Definition 2). The authors correctly state that any environment can be decomposed this way, as the errors can absorb any discrepancy. However, the utility of the bound in Theorem 2 depends on these errors being small.

- **Dependency Scores for the Heuristic**: The practical heuristic (Algorithm 1) requires pre-defined "dependency scores" to guide the merging process. In all experiments, these scores are based on domain-specific spatial proximity (e.g., Euclidean or Manhattan distance, or binary state position) . This seems to be a form of privileged information.

- ** Strictness of Assumption 3(iii)**: The theoretical framework (specifically Assumption 3(iii)) requires no edges between subteams. This is noted as a technical requirement for the proof. This hard partitioning seems somewhat rigid, as real-world teams often have sparse but important cross-team coordination.

- **"Line" DAG Baseline**: In the tabular results (Section 6.1), the "line" DAG is used as a baseline and performs quite well (Figure 1). The paper notes this DAG does not satisfy Assumption 3  (as it's a single chain, not a partition of fully-connected subteams). This is an interesting result.

**Questions:**

- (W1) Could the authors comment on the generality of low-error decomposability? The tabular experiments show this holds for the Coordination Game, but how restrictive is this assumption for more complex, non-obvious benchmarks? Is it possible to estimate or bound these errors a priori to determine if an environment is suitable for this method?

- (W2) How sensitive is the heuristic to the quality of these scores? What would happen in a task where the primary dependency is functional (e.g., "passer" and "shooter") rather than spatial? Could these dependency scores be learned dynamically, perhaps by analyzing value function contributions or action correlations, to make the approach more general?

- (W3) How challenging would it be to relax this assumption? For example, if a small number of inter-team edges were permitted, would this completely break the value function decomposition (Lemma 5), or could it be handled by introducing additional, bounded error terms into the analysis?

- (W4) Does this suggest that the 'fully-correlated subteam' structure is a sufficient, but perhaps not strictly necessary, condition for good performance? It implies that other sparse, structured DAGs might also be effective, even if they don't fit the specific subteam partition model.

---

> ### Author Response · Authors · 2025-11-26
>
> Thank you for carefully reviewing our paper!
> Please find below our response to the Weaknesses (W) and Questions (Q).
>
>
> **(W1, Q1) The "decomposability" assumption**
>
> We agree the bound is only meaningful when errors are small; below we clarify when that happens and how one might estimate it.
>
> One can always achieve smaller decomposition errors with fewer subteams, with the extreme case being $K=1$ subteam with zero errors. For $K>1$, quantitatively measuring the decomposition error is difficult without explicit access to the transition and reward functions, and even with full knowledge it may be computationally intractable to compute exactly. Nonetheless, many cooperative MARL environments exhibit approximately local interactions that naturally lead to low decomposition error under reasonable partitions. For example, in Predator–Prey, when predators are effectively chasing different preys that are far apart, the two groups have minimal interaction and the induced decomposition error is expected to be small.
>
> If $P$ and $r$ are known, we can estimate errors $|\epsilon_P|$,$|\epsilon_r|$ by fitting factorized transition/reward models as in Table 1 and Appendix B.2. This can yield a suitability diagnostic.
>
>
> **(W2, Q2) The dependency scores**
>
> For Predator–Prey, Warehouse, and Battle,  the dependency scores are based on physical (Euclidean) distances.
> For Coordination Game and Aloha, we would say the dependency scores are “functional”:   in the Coordination Game, the score is not based on physical distance; instead, it is derived directly from the agents’ local binary states; In Aloha, the agents are arranged in a 2d grid that captures whether two agents would functionally interfere with each other.
>
> Indeed, these dependency scores are based on domain knowledge and do not involve any learning. We agree that the approach would be more general by leveraging learned dependencies.
>
> We would not say the dependency scores are of “high quality”, as they are completely general purpose and not particularly designed for reducing decomposition errors. Therefore, we were surprised that, with such simple dependency scores and heuristics, our method outperforms the best learning-based variant of VAST (even at convergence). This suggests that such learning-based methods still have quite room for improvement.
> We agree a learned score function would further generalize the method.
>
>
>
> **(W3, Q3) Assumption 3(iii)**
>
> This condition is indeed critical in the proofs: e.g., a key step in Lemma 3’s proof is to upper bound the gain of $a^{\mathcal C_k}$ over the subteam baseline $V_\theta^{k}(s)$, which derived from a telescoping sum that would fail if subteams are not independent (see Line 1050 in Appendix A.5 for details).
> Therefore, allowing a small number of inter-team edges would break the exact decomposition used in Lemma 5. Intuitively, these cross-team edges would introduce additional interaction terms that could likely be treated as extra bounded errors, in the same spirit as the decomposition errors already handled in Theorem 2. However, working out the precise bounds and ensuring a clean convergence analysis appears nontrivial, and we do not yet know whether this can be done in a fully general way.
>
>
> **(W4, Q4) The 'fully-correlated subteam' structure, the "Line" DAG baseline**
>
> The 'fully-correlated subteam' structure is necessary for the theoretical guarantees, since, without full correlation, algorithms can converge to suboptimal policies in general.
> In practice, we agree that certain sparse DAGs without full correlation can also be effective, as shown by our “line” DAG empirical results.
> Specifically, the line DAG can be understood in two views:
> - It is a sparse version of a single fully-correlated N-agent subteam.
> - It is N subteams, each with a single agent (hence fully-correlated), with inter-subteam edges (i.e., violating Assumption 3(iii)).
> Both views relax our theoretical assumptions.

---

### Meta-Review · Area_Chair_sTid · 2026-01-02

**Summary:**

In this paper, the authors consider the extended policy architecture for multi-agent RL. Conventionally, for MARL, the product joint policy has been adopted, i.e., the joint policy is the product of individual polices. But, the authors went beyond this conditionally independent policy distribution to consider directed acyclic graph-based Bayesian network policy architecture. Under the soft-max parameterization, they showed that the finite-time convergence of BN policies under the subteam decomposition under several assumptions.  The work is considered meaningful in terms of MARL theory.

**Reviewer Concerns:**

The theory part seems OK but the empirical part of the paper is rather weak.
Several assumptions used in proof may not hold in real experiment setup.
Performance gain seems marginal.

AC comments:

As stated above, this paper is meaningful in terms of MARL theory. But, the proposed BN policy architecture requires communication among the members of a subteam and sequential action generation is required. The whole benefit of product policies is to circumvent these difficulties and to enable fullly decentralized operation at test time.  It is recommended that the authors provide some example environment on which the propose method yields true gain over product polices. The current baselines are too weak. There exist many strong baselines now and comparison with these strong baselines are required too to quantify how much gain one can truely get from this complicated policy modeling.

**Reviewer Scores:**

Reviewer Qy1s 8,   Reviewer 2vBZ 6,  Reviewer Duum 6, Reviewer g5cJ 8

Due to the contribution to MARL theory, it is expected that the reviewers will keep their scores, although there exists some weakness in experiments.

---

### Decision · Program_Chairs · 2026-01-26

Accept (Poster)